# Comparative Metabolomics Profiling Reveals Key Metabolites and Associated Pathways Regulating Tuber Dormancy in White Yam (*Dioscorea rotundata* Poir.)

**DOI:** 10.3390/metabo13050610

**Published:** 2023-04-28

**Authors:** Jeremiah S. Nwogha, Abtew G. Wosene, Muthurajan Raveendran, Jude E. Obidiegwu, Happiness O. Oselebe, Rohit Kambale, Cynthia A. Chilaka, Veera Ranjani Rajagopalan

**Affiliations:** 1Department of Horticulture and Plant Sciences, College of Agriculture and Veterinary Medicine, Jimma University, Jimma P.O. Box 307, Ethiopiawosene.gebreselassie@ju.edu.et (A.G.W.); 2Centre for Plant Molecular Biology & Biotechnology, Departments of Plant Biotechnology and Biochemistry, Tamil Nadu Agricultural University, Coimbatore 641003, India; raveendrantnau@gmail.com (M.R.); rohitkamble568@gmail.com (R.K.);; 3Yam Research Programme, National Root Crops Research Institute, Umudike 440001, Nigeria; ejikemeobidiegwu@gmail.com; 4Department of Crop Production and Landscape Management, Ebonyi State University, Abakaliki 480282, Nigeria; h.oselebe@gmail.com; 5Institute for Global Food Security, School of Biological Sciences, Queen’s University Belfast, 19 Chlorine Gardens, Belfast BT9 5DL, UK

**Keywords:** metabolites, differential–accumulation, energy, antioxidant, metabolism, molecular–mechanism, dormancy, regulation

## Abstract

Yams are economic and medicinal crops with a long growth cycle, spanning between 9–11 months due to their prolonged tuber dormancy. Tuber dormancy has constituted a major constraint in yam production and genetic improvement. In this study, we performed non-targeted comparative metabolomic profiling of tubers of two white yam genotypes, (*Obiaoturugo* and *TDr1100873*), to identify metabolites and associated pathways that regulate yam tuber dormancy using gas chromatography–mass spectrometry (GC–MS). Yam tubers were sampled between 42 days after physiological maturity (DAPM) till tuber sprouting. The sampling points include 42-DAPM, 56-DAPM, 87DAPM, 101-DAPM, 115-DAPM, and 143-DAPM. A total of 949 metabolites were annotated, 559 in *TDr1100873* and 390 in *Obiaoturugo*. A total of 39 differentially accumulated metabolites (DAMs) were identified across the studied tuber dormancy stages in the two genotypes. A total of 27 DAMs were conserved between the two genotypes, whereas 5 DAMs were unique in the tubers of *TDr1100873* and 7 DAMs were in the tubers of *Obiaoturugo*. The differentially accumulated metabolites (DAMs) spread across 14 major functional chemical groups. Amines and biogenic polyamines, amino acids and derivatives, alcohols, flavonoids, alkaloids, phenols, esters, coumarins, and phytohormone positively regulated yam tuber dormancy induction and maintenance, whereas fatty acids, lipids, nucleotides, carboxylic acids, sugars, terpenoids, benzoquinones, and benzene derivatives positively regulated dormancy breaking and sprouting in tubers of both yam genotypes. Metabolite set enrichment analysis (MSEA) revealed that 12 metabolisms were significantly enriched during yam tuber dormancy stages. Metabolic pathway topology analysis further revealed that six metabolic pathways (linoleic acid metabolic pathway, phenylalanine metabolic pathway, galactose metabolic pathway, starch and sucrose metabolic pathway, alanine-aspartate-glutamine metabolic pathways, and purine metabolic pathway) exerted significant impact on yam tuber dormancy regulation. This result provides vital insights into molecular mechanisms regulating yam tuber dormancy.

## 1. Introduction

Yam is a multi-species crop that has about 613 known species that produce tubers, bulbils, or rhizomes [1]. Of these, about 10 are cultivated over a larger area and serve as a staple food crop, and about 50 other species are also eaten as wild-harvested staple famine food; thus, this genus occupies a prominent position in global food insecurity combat [2,3]. Yam is placed fourth position among the utilized root and tuber crops globally, after potatoes (*Solanum* spp.), cassava (*Manihot esculenta*), and sweet potatoes (*Ipomoea* spp.), and second in West Africa after cassava [4,5]. However, the sensorial preference for yam coupled with its better organoleptic properties and storability compared to cassava, potatoes, sweet potatoes, and plantain has led to its high demand as a major cash crop in Sub-Sahara Africa [1]. Its potential as a source of food is attributed to its high levels of carbohydrates including fiber, starch, and sugar, contributing about 200 dietary calories per person per day to more than 300 million people in the tropics [6], in addition to other nutritional benefits such as proteins, lipids, vitamins, and minerals [2]. Yam tuber also contains an abundance of steroidal C_27_ saponins, and diosgenin. the aglycone portion of the abundant saponin dioscin has been industrially exploited as the starting material for the synthesis of pregnenolone-derived steroids, thus making yam an industrial crop [7].

Yam tuber dormancy is the temporary growth arrest of the continuous underlying meristematic cells beneath the dead brown skin of the tuber [8]. It is an inherent mechanism that regulates tuber sprouting. The dormancy phase has been divided into three categories according to Lang, et al. [8], including: endodormancy (deep dormancy during which growth arrest is influenced by internal physiological and genetic factors within the meristem), para-dormancy (this occurs when growth is arrested by physiological factors external to the meristem), and eco-dormancy (growth is stopped by unfavorable external or environmental factors).

Prolonged tuber dormancy after physiological maturity has constituted a great constraint in yam crop production and genetic improvement [9]. Tuber dormancy is the major cause of ware and seed tubers’ inability to sprout for a prolonged period, during which tubers remain dormant; incapable of developing internal or external shoot buds/sprouts for about 150 to 210 days depending on: species, genotype, and growing and storage environmental conditions [10,11,12]. Thus, this makes it impossible to have more than one crop cycle per year, thereby limiting yam crop productivity, tuber availability for industrial use, and slowing down the rate of genetic improvement [13]. Dormancy induction and release is associated with numerous physiological and biochemical processes that are regulated by gene expression, protein synthesis, hormonal signaling, and energy metabolism [14]. In these processes, both the decomposition and synthesis of metabolites, conversion, and consumption of energy occur, thus generating a genotype and dormancy stage-specific metabolites [15,16]. Some of these metabolites have been found to regulate dormancy in seeds of various plant species. The typical example is phytohormones, whose involvement in seed dormancy regulation has always been a popular topic in seed physiology and biochemistry research [17]. The most reported phytohormone influencing dormancy induction in crops is abscisic acid (ABA), which inhibits the process of seed germination, though complex interactions exist between ABA and other phytohormones [18]. Gibberellic acid (GA) is another important phytohormone known to be involved in dormancy regulation. It acts antagonistically with ABA to promote seed germination (dormancy breaking), while jasmonic acid (JA) shows a synergetic effect with ABA in regulating dormancy induction and maintenance in plants [7]. Exogenous treatment with auxin in soybean was found to induce dormancy by decreasing the GA/ABA ratio [19]. Therefore, the roles of these phytohormones in dormancy regulation are complex and subtle.

Polyamine has also been reported to play a key role as a modulator of plant growth and development, and its involvement in seed dormancy regulation has attracted research attention recently. Dormancy-breaking led to the accumulation of the total polyamine content in the seeds [20]. Polyamine was found to be distributed compartmentally in dormancy-breaking seeds, which may suggest its involvement in the process [21]; presoaking with polyamine improved seed germinability [22]. This effect has been suspected to involve starch metabolism, phytohormones interactions, and antioxidant defenses [23,24]. In addition, the involvement of amino acids in dormancy regulation has also attracted research attention. For instance, it has been reported that essential amino acids such as lysin, methionine, leucine, isoleucine, threonine, phenylalanine, and valine increased significantly during the dormancy breaking process in wheat, brown rice, triticale, and Arabidopsis [25,26]. Asparagine, arginine, and γ-aminobutyric (GABA) are the three most common amino compounds in the mobilization of nitrogen reserves in dormancy breaking chestnut seeds [27]. Therefore, as the major transporter of all forms of nitrogen and the key factor of carbon/nitrogen balance in the plants, the role of amino acids in dormancy regulation should not be ignored [18].

Carbohydrates play multiple roles in plant growth and development, besides acting as the primary source of carbon and energy; hence, energy metabolism has also been implicated in dormancy regulation. It has been reported that Embden Meyerhof Parnas (EMP) pathway, tricarboxylic acid (TCA) pathway, and pentose phosphate pathway (PPP) strictly regulated dormancy in different plants [28]. For instance, the TCA cycle was enhanced while the PPP pathway was slowly decreased during apple bud dormancy breaking [29]. In grape, dormancy breaking was induced by chemicals and low temperature was regulated by PPP, EMP, and TCA cycles [30,31]. Also, other carbohydrates generating energy pathways has been reported to be involved in dormancy regulation in herbaceous plants; for example, sucrose metabolism, lipid metabolism, and amino acid metabolism were all downregulated in imbibed dormant seeds of herbaceous plants [32], thus demonstrating that the ability of imbibed seeds to synthesize protein and generate adenosine triphosphate (ATP) decreases during the dormancy period [33,34]. Also, two studies have reported that the imbibed dormant seeds of *Picea glauca* and *Julans regia* exhibited an inactivation of sucrose and amino acids due to wet cold induced dormancy [32]. Furthermore, an accumulation the galactinol and raffinose family of oligosaccharides (RFOs) during seed maturation stages in some legumes has indicated their roles in desiccation tolerance and longevity [35]. Galactinol content in dry seeds has also been used as a biomarker for seed longevity in *Brassiceae* and tomato [36]. The sucrose-non-fermenting 1-related kinase1 (SnRK1) transcription family has been reported to play the role of energy metabolism sensors in plant systems; they detect when energy is below optimum in the plant system and inhibit plant growth and developmental processes that are energy-consuming, such as dormancy-breaking (sprouting), in order to maintain energy homeostasis during a low-energy economy, which characterizes the dormant state [37,38]. There are also reports that trehalose-6-phospate (T6P) initiates the process of growth in any plant part undergoing growth arrest (dormancy) by inhibiting the action of (SnRK1) [39,40].

The effective reduction of yam tuber dormancy duration through genetic manipulation is considered a reliable solution to the challenge of prolonged tuber dormancy and ensures more than one yam crop cycle per annual. It is important to note that the paucity of genetic resource information on factors controlling the dormancy trait has limited efforts towards overcoming the barriers mentioned above. Our study presents an alternative route in understanding implicated metabolites and their associated pathways at the biochemical and genetic level which could be utilized in genetic manipulation yam tuber dormancy.

The development of metabolomics technologies has improved our understanding of the complex molecular interaction of biological systems. Due to its high throughput and efficiency, metabolomics has been applied to research on the mechanism of dormancy in several plant species [18]. However, in yam crop only few metabolomics studies exist, and to the best of our knowledge this is the first time in which the metabolomics approach is being applied to study yam tuber dormancy. In this study, an untargeted GC-MS approach was used to profile tubers of two white yam genotypes (*Obiaoturugo* and *TDr1100873*) for differentially accumulated metabolites from dormancy induction to dormancy breaking in order to determine the metabolites and their associated pathways that regulate yam tuber dormancy.

## 2. Experimental Design

### 2.1. Genetic Material

Two yam genotypes (*Obiaoturugo* and *TDr1100873*) of white yam specie (*D. rotundata,*) were used for the study. *Obiaoturugo* is a popular landrace while *TDr1100873* is an advanced breeding clone. Both individuals were sourced from the Yam breeding program of the National Root Crops Research Institute (NRCRI), Umudike, Nigeria.

### 2.2. Field Study Area

The field was established at an Eastern research farm of the National Root Crops Research Institute (NRCRI) in Umudike, Nigeria. Umudike is a rainforest agroecology located at; 5.4729° N, 7.5480° E, 152 m ASL with a mean annual rainfall of 2093 mm, mean annual temperature of 27.3 °C, mean annual relative humidity of 82%, and sandy loam soil with a pH range of 4.3 to 5.27.

### 2.3. Planting and Agronomic Management

A uniform set size (200 g) of only proximal and distal regions of the tubers was planted in order to maintain relatively uniform germination time. A total of 30 plant stands of each genotype were planted in randomized complete block design, replicated three times, with 10 plant stands of each genotype per replicate. The field was adequately maintained under a rainfed condition, and all cultural practices including application of NPK fertilizer at the recommended dose of 80:60:100 kg/ha [41] were carried out. Tubers were harvested at 50% senescence, which is the tuber physiological maturity stage. The Figure 1 below shows the vegetative growth and tuber storage stages.

### 2.4. Postharvest Study and Sampling

After harvesting, tubers were freighted to the Centre of Plant Molecular Biology and Bioinformatics, Department of Plant Biotechnology Tamil Nadu Agricultural University, Coimbatore, India, where the tubers were stored in a greenhouse facility with natural light conditions and sunshine hours per day between 10 to 12 h, a relative humidity range of 38% to 78%, and a maximum and minimum temperature range from 26 to 36.5 °C and 17.5 to 26.5 °C, respectively, during the study. Triplicate samples of tubers were collected at the following days after physiological maturity (DAPM): 42-DAPM, 56-DAPM, 87-DAPM, 101-DAPM, 115-DAPM and sprouted tuber (143-DAPM). The two genotypes (*Obiaoturugo* and *TDr1100873*) exhibited different dormant phenotypes. *Obiaoturugo* exhibited a shorter dormant phenotype and sprouted at 101 days after tuber physiological maturity, and was sampled only four times within this period at 42, 56, 87, and 101 days. On the other hand, *TDr1100873* exhibited longer dormant phenotype and sprouted at 143 days after tuber physiological maturity, and was sampled six times within the period at 42, 56, 87, 101, 115, and 143 days after tuber physiological maturity. The samples were frozen in liquid nitrogen immediately after collection, and stored at −80 °C until the sampling was completed. At the completion of sampling, freeze-dried samples were microwaved for 5 min and then oven-dried in an oven at 72 °C for 48 h, and grinded into powder using a commercial grinder.

### 2.5. Metabolites Extraction

A total of 10 g of powdered samples were wrapped in filter papers, and dropped in the cylindrical tubes of Soxhlet apparatus and filled with 400 mL of 100% methanol solvent produced by Sigma–Aldrich. The temperature of the Soxhlet apparatus was set to 70 °C. Each sample was subjected to nine extraction cycles, and 50 mL of the extract were collected in 50 mL centrifuging tubes at the end of the nine extraction cycles, then air-dried to 7.5 mL before further processing.

#### Sample Processing

A total of 500 µL of the extracts was pipetted into 2.0 Eppendoff tubes and incubated in a water bath for 15 min at 37 °C, before centrifuging at 10,000 rpm and 4 °C for 10 min. Samples were filtered using 0.2 mm filter paper and syringe. After filtering, samples were diluted with methanol at a ratio of 1:9. A total of 500 mL of the diluted samples were concentrated at 20,000 rpm for 2 h. The concentrated solid extracts were derivatized according to [1] with slight modifications by dissolving it in 50 mL solution of (20 mg methoxylamine hydrochloride + 1 mL of pyrimidine), and incubated for 1.5 h at 37 °C. Afterward, 80 µL of N-Methyl-N-(trimethylsily)trifluoroacetamide was also added and incubated for 30 min at 37 °C. The derivatization chemicals were sourced from Sigma–Aldrich (Bengaluru, Karnataka 560099, India).

### 2.6. GC-MS Analysis

Untargeted metabolomics data were acquired using two auto-injecting GC-MS machines (Turbomass and 8040NX) by injecting 1ml of each sample. The GC oven was held for 2 min at 705 °C, ramp for 10 °C/min to 150 °C, then held for 2 min, ramp at 5 °C/min to 220 °C, held for 1 min, ramp 10 °C/min to 250 °C, held for 1 min. Auto-injection was conducted at 220 °C. Helium was the carrier gas at a flow rate of 1.3 mL/min. The interface with the MS was set at 290 °C and MS performed in full scan mode using 70 eV EI + and scanned from 50 to 500 Da. Retention time locking to ribitol was used according to Enfissi, et al. [42]. A mixture of n-alkanes, ranging from 8 to 32 carbons, was used for retention index external calibration. One milliliter of the extraction solvent (methanol) was used as the internal standard. Sample sets were run in batches of three replicates, Solvent Delay = 3.00 min, Transfer Temp = 250 °C, Source Temp = 220 °C, Scan mass range: 50 to 500 Da. Column was set at 30.0 m × 250 μm.

#### Data Processing

To identify chromatogram components found in white yam tuber profiles, a mass spectral library was constructed from machine in-built libraries: NIST ‘20M&R MS lib, Main lib, rep lib, OA_TMS_DB5_67min_V3 lib and FA_ME_SP2560_V3 lib. Component peak identification and spectral deconvolution was performed using the Automated Mass Spectral Deconvolution and Identification System (AMDIS v2.71, NIST) using Kovat’s retention indices (RI) and MS for identification according to the metabolomics reporting guidelines [1]. Each compound was assigned a representative ion and response areas were integrated and expressed relative to internal standard. Linearity was assessed following the standard extraction procedure on amounts of material (10 mg) and the range further enhanced by taking the quantity of aliquot used (400 μL). Recovery was assessed by expressing measurements from each extract of sample (10 mg) as a percentage of the total following three sequential extractions. All measurements for the developmental stages were conducted in triplicate and relative to internal standard. 

### 2.7. Data Statistical Analysis and Visualization

Data were further processed in metaboAnalyst 5.0 platform. Row-wise normalization by the sum, logarithmic transformation and scaling by mean centering were performed to normalize the data (Appendix A). All statistical analyses were perform using metaboanalyst software. Venn tool (Version 2.1) [43] was used to determine the relationship between differential accumulated metabolites (DAMs) in the tubers of the two genotypes. One-way univariate ANOVA analyses were performed to determine the significant differentially accumulated metabolites across different data points at a *p*-value threshold of *p* ≤ 0.05, and Turkey’s HSD was used to compare the means. Principal component analysis (PCA) was performed to determine the overall pattern of metabolites distribution in the two genotypes along the data points. In order to filter out the variables in the metabolites that were not related to categorical variables and to obtain more reliable metabolite information on the structure of different data points, partial least squares–discriminant analysis (PLS–DA) was performed using variable importance in projection (VIP) at a threshold of VIP ≤ 1. Agglomerative hierarchical cluster analysis was performed to classify the differentially accumulated metabolites (DAMs) across the data point using euclidean and ward.D distance algorithms and visualized it in a clustering heatmap. The DAMs were mapped into PubChem, Kyoto encyclopedia genes and genomes (KEGG) and human metabolome data base (HMDB) to identify their functional groups. Metabolites set enrichment analysis (MSEA) and pathway annotation were performed to determine the significantly induced metabolisms that DAMs are involved in, during the studied yam dormancy stages. Pathway topology analysis was performed to visualize the significantly induced metabolic pathways.

## 3. Results

### 3.1. Metabolic Profiling of White Yam Tuber during Dormancy

A total of 949 metabolites were annotated in tubers of the two genotypes during the study period; 390 metabolites were annotated in the tubers of *Obiaoturugo* and 559 metabolites annotated in the tubers of *TDr1100873*. Using Venn interactive tool (accessed on 1 November 2022) [43], we compared the tuber metabolomes of the two genotypes and it showed that 158 metabolites were common between the genotypes (Figure 2a). Differentially accumulated metabolites (DAMs) in tubers of the two genotypes were also compared, and Figure 2b showed that 27 DAMs were conserved in the tubers of both genotypes, while 7 unique DAMs were observed *Obiaoturugo* and 5 unique DAMs were observed *TDr1100873*. It is believed that the phenotypic variation between the two yam genotypes with respect to dormancy duration was driven by these unique DAMs.

Analysis of variance (ANOVA) with Bonferroni correction identified 28 and 30 significantly differentially accumulated metabolites at (*p* ≤ 0.05 and FDR ≤ 0.05) across the data points in the tubers of *Obiaoturugo* and *TDr1100873* respectively (Figure 3a,b).

### 3.2. Principal Component Analysis (PCA) and Partial Least Square–Discriminant Analysis (PLS–DA) of Metabolites

Principal component analysis (PCA) was performed to determine the overview of yam tuber metabolome at the six data points of *TDr1100873* and four data points of *Obiaoturugo*. Figure 4a shows the discriminating pattern of *TDr1100873* tuber metabolome across the studied dormancy stages. The first principal component (PCI) clearly separated *TDr1100873* tuber metabolome at 42-DAPM, 115-DAPM and 143-DAPM separated from the rest of the data points and it accounts for 28.4% of total variation, while PC2 accounts for 21.4% of the total variation and it drives the clustering together of the tuber metabolome at 56-DAPM, 87-DAPM, and 101-DAPM. This indicates that the later three dormancy stages shared similar metabolome, while the former three dormancy stages were distinct. Although 115-DAPM and 143-DAPM together constitute the dormancy-breaking stage and were expected to share similarities in their metabolomes, PCA did not detect any similarity between them. The tuber metabolome at 115-DAPM, which marks the appearance shoot buds (the first physical sign that tuber is getting ready for sprouting), differed substantially from the metabolome at 143-DAPM (sprouted tuber). We used PCA loading to identify the major metabolites contributing to the tuber metabolome variations at these dormancy stages (Appendix A). The metabolites playing key roles in tuber metabolome discrimination at different dormancy stages include: four esters, four amines, two alcohols, an alkaloid, fatty acid, lipid, coumarin, terpenoid, and one flavonoid. Similarly, Figure 4b shows an *Obiaoturugo* tuber metabolome discriminating pattern across its four studied dormancy stages. The tuber metabolomes at 87-DAPM and 101-DAPM were clearly separated from that at 42-DAPM and 56-DAPM, respectively, by PC1, and it accounted for 50.5% total variations. PC2 accounts for 32.4% of total variations and drives the clustering together of 42-DAPM and 56-DAPM together. This is an indication that *Obiaoturugo* tuber metabolome at these dormancy stages are similar, implying that during the 14 days between the two sampling points no substantive metabolic change occurred. However, the distance between the tuber metabolome at 87-DAPM and 101-DAPM indicates that tuber metabolomes at these two stages are clearly different from each other. 87-DAPM marked the appearance of shoot buds in the tubers of *Obiaoturugo*, while 101-DAPM is its sprouted tuber metabolome. PCA loading revealed that the metabolites driving the discrimination of *Obiaoturugo* tuber metabolome at 87-DAPM and 101-DAPM were: five esters, three amines, three alcohols, lipid, fatty acid, coumarin, flavonoid, and a carbohydrate (Appendix A). 

We performed partial least squares–discriminant analysis (PLS–DA) to further access the performance of the discrimination of the tuber metabolome under a supervised technique across the studied stages, using the very importance in projection (VIP) scores to evaluate the pattern of differential accumulation of important metabolites identified by PLS–DA across the dormancy stages (Appendix A). Figure 4c shows the discriminating of *TDr110873* tuber metabolome under supervised classification technique across the investigated dormancy stages. Component 2 separated tuber metabolome at 115-DAPM and 143-DAPM from the metabolome in the rest of the dormancy stages, and it accounts for 16.8% of the total variations, while component 1, which accounts for 15.4% variation, separated the tuber metabolome at 101-DAPM from 42-DAPM, 56-DAPM, and 87-DAPM. Figure 4d shows the discriminating pattern of *Obiaoturugo* tuber metabolome across the studied dormancy stages under supervised classification. It indicates that the *Obiaoturugo* metabolome at each stage represents a distinct dormancy progression stage unlike what was observed with PCA. The separations were driven by components 1 and 2 of the PLS–DA scores plot which accounts for 32.6% and 36.9% of the total variations, respectively. The studied dormancy stages exhibited a consistent pattern in their separation into groups of clusters. The separation into distinct metabolome group or clustering into groups of similar metabolomes was determined mainly by the time duration between dormancy stages and dormancy phenotype, which necessitate change in metabolome composition.

### 3.3. Identification of Differentially Accumulated Metabolites across the Studied Tuber Dormancy Stages

To identify the differential accumulated metabolites (DAMs) across tuber dormancy stages studied, we performed agglomerative hierarchical clustering (AHC) analysis which partitioned DAMs earlier identified by ANOVA and PLS–DA VIP scores into the pattern of the exact change in their concentrations across the different dormancy stages using Euclidean distance measure and ward. D clustering algorithm. If a metabolite plays a role in the biological process, its concentration normally fluctuates during the process to adapt to the physiological changes in the process. First, AHC grouped the studied dormancy stages of *TDr1100873* tuber into two main clusters; cluster 1 comprises of 143-DAPM, 115DAPM, and 42-DAPM, cluster 2 comprises of 101-DAPM, 87-DAPM, and 56-DAPM (Figure 5a). In *Obiaoturugo*, the only distinct cluster comprises of 56-DAPM and 87-DAPM, while 42-DAPM and 101-DAPM were separately linked to it (Figure 5b). This was a deviation from the normal three curve stages in the germination process of botanic seed. However, this was expected because the physical characteristics of dormant botanic seeds that influence its growth curve exist in yam tuber. While botanical seeds are in the desiccation state at maturity, necessitating their requirement of imbibition during the germination process, yam tubers at maturity are in the hydrated state, and do not require imbibition during germination. Instead, as we reported earlier [44], it requires moisture loss to a 40% threshold before the dormancy-breaking process can commence. This might be linked to this sharp deviation of yam tuber germination stages from well-established three curve stages of the germination process in botanical seeds.

Agglomerative hierarchical clustering of the DAMs revealed the dynamic changes in the tuber metabolomes of the two genotypes (*TDr1100873 and Obiaoturugo*) across the studied dormancy stages. It identified 32 DAMs in tubers of *TDr1100873*, and 34 DAMs in tubers of *Obiaoturugo* (Figure 5a,b). Figure 5a shows that in *TDr1100873*, eight DAMs which comprised of aromatic amine, (two) alcohols and polyols, organosiloxane, amino acid and derivatives, alkaloid, biogenic polyamine, coumarin were upregulated, and 15 DAMs that comprised of nucleotide (adenine), (two) alkaloids, amino acid and derivative, galactonolactone (sugar acid), flavonoids, fatty acid, (three) carboxylic acids, coumarin, (two) alcohols and polyols, ester, benzene derivatives and phytohormone, were downregulated at 42-DAPM. At 56-DAPM, nine DAMs which comprised of aromatic amine, (two) amino acids and derivatives, (two) alcohols, flavonoid, organosiloxane, biogenic polyamines and ester were up-regulated, and seven DAMs that comprised of benzene derivatives, phenol, sugar (sucrose), benzoquinone, alcohol and polyol, ester, and carboxylic acid were down-regulated. At 87-DAPM, 10 DAMs which comprised of fatty ester, biogenic polyamine, flavonoid, amines, alcohol, phenol, lipid, coumarin, fatty acid, benzene derivatives were up-regulated, and seven DAMs that comprised of, alkaloids, (two) nucleotides (adenine and nucleic acid), aromatic amine, alcohol and polyol, amino acid and derivatives, organosiloxane were down-regulated. At 101-DAPM, six DAMs which comprised of fatty ester, terpene, alcohol, (two) nucleotides (adenine and nucleotide sugar acid) and alkaloid, were up-regulated, alkaloids, and seven DAMs that comprised of alcohol, flavonoid, amine, biogenic polyamine, alkaloid, phenol, phytohormone (salicyluric acid) down-regulated. At 115-DAPM, seven DAMs which comprised of sugar (sucrose), benzoquinone, (two) carboxylic acids, fatty acid, lipid, phytohormone (salicyluric acid) were up-regulated, and 12 DAMs that comprised of fatty ester, biogenic polyamine, terpene, (two) amino acid and derivatives, (three) alcohols and polyols, (two) flavonoids, fatty acid, organosiloxane were down-regulated. At 143-DAPM, 12DAMs which comprised of (two) flavonoids, sugar (sucrose), (two) alkaloids, carboxylic acid, phytohormone (salicyluric acid), biogenic Polyamines, benzene derivatives, amino acid, benzoquinone were up-regulated, whereas seven DAMs that comprised of aromatic amine, alcohol, amino acids and derivatives, sugar acid, flavonoid, fatty acid, lipid were down-regulated. 

Figure 5b shows that in *Obiaoturugo*, 15 DAMs which comprised of (three) amino acids and derivatives, (three) alcohols and polyols, (two) alkaloids, (two) carboxylic acids, (two) phytohormones (salicyluric and 3-methylsalicylic acids), sugar acid, amino alcohol, biogenic polyamine were up-regulated, and 14 DAMs that comprised of (two) carboxylic acids, (two) esters, (two) flavonoids, phenol, fatty acid, aromatic amine, organosiloxane, benzoquinone, down-regulated at 42-DAPM. At 56-DAPM, seven DAMs which comprised of amine, (two) flavonoids, benzene derivatives, (two) nucleotides (adenine and nucleic acid), sugar (sucrose) were up-regulated, and five DAMs that comprised of ester, nucleotide (nucleotide sugar), alcohol, biogenic polyamine, and alkaloid were down-regulated. At 87-DAPM, 11 DAMs which comprised of (two) nucleotides (nucleic acid and nucleotide sugar), benzoquinone, (two) esters, aromatic amine, fatty acid, carboxylic acid, flavonoid, alcohol, organosiloxane were up-regulated, and four DAMs that comprised of flavonoids, phytohormone (methylsalicylic acid), alkaloids, amino alcohol were down-regulated. At 101-DAPM, eight DAMs which comprised of (two) flavonoids, benzoquinone, ester, carboxylic acid, alcohol organosiloxane and biogenic polyamines were up-regulated, and nine DAMs that comprised of phytohormone (methysalicylic acid), amino acid and derivatives, alcohols and polyols, sugar acid, inorganic oxide, (two) carboxylic acids were down-regulated. This result demonstrates that amines, biogenic polyamines, amino acids and derivatives, alcohols, flavonoids, alkaloids, organosiloxane, phenols, esters, coumarins and phytohormone dominated the metabolic activities during the dormancy period, whereas, fatty acids, lipids, nucleotides, carboxylic acids, sugars, amino acid (glutamine), terpenoids, benzoquinones, and benzene derivatives dominated the metabolic activities during tuber dormancy breaking and sprouted tubers of both yam genotypes. However, polyamine accumulated in the tubers of both genotypes during the deep dormant stages and in sprouted tubers, indicating that it might be playing a role in both dormancy induction and yam vine development. 

### 3.4. Metabolites Mapping and Chemical Functional Groups Identification

To perform further analysis with the identified DAMs the compound labels, or identity need to be standardized. The raw metabolites labels were used as queries in mapping DAMs into HMDB, PubChem, and KEGG databases to identify their standardized compound names, chemical IDs, and functional groups. It was observed that the DAMs were distributed across sixteen (13) major chemical functional groups, comprising primary and secondary metabolisms. Alcohols and polyols were the most abundant DAMs, and it constitutes 12.82% proportion of the total DAMs, followed by fatty acids and esters, and amines and polyamines as each of them constitute 10.26% (Figure 6). Table 1 shows the conversion of the raw DAMs labels to standard compound names, their chemical functional groups, and IDs in HMDB, PubChem, and KEGG. 

### 3.5. Metabolites Sets Enrichment Analysis and Pathways Annotation

We performed metabolite set enrichment analysis using a hypergeometric test to determine whether the metabolism of any set of DAMs is represented more than expected by chance during the studied yam tuber dormancy stages. It was found that 12 metabolisms were significantly enriched at (*p* ≤ 0.05) during the studied yam tuber dormancy stages (Figure 7). The enrichment ratio of each of the induced metabolism denoted by the length of its bar length indicates the fold change in a particular metabolism from normal, while the *p*-value indicates determines whether a particular metabolism has a significant impact on the biological process under consideration. Linoleic acid metabolism recorded the highest enrichment ratio of over 20 at *p* = 0.02, an indication that linoleic acid metabolism plays the most important role in yam tuber dormancy regulation. On the other hand, other metabolisms, such as galactose metabolism, biosynthesis of unsaturated fatty acids, pyrimidine metabolism, nitrogen metabolism, purine metabolism, D-glutamine and D-glutamate metabolism, phenylalanine metabolism, alpha-linoleic acid metabolism, Arginine biosynthesis, Glycerolipid metabolism, and starch and sucrose metabolism play some roles in yam tuber dormancy regulation.

### 3.6. Pathway Topology Analysis

Since the metabolic network is directed, we used the relative betweenness centrality of the metabolites in a node to determine the impact of the pathway on yam tuber dormancy regulation. The result indicates that seven metabolic pathways (linoleic acid metabolic pathway, phenylalanine metabolic pathway, galactose metabolic pathway, starch and sucrose metabolic, alanine, aspartate, and glutamate metabolic pathway, purine metabolic pathway, and lysine biosynthesis) had a significant impact on yam tuber dormancy regulation, with linoleic acid metabolic pathway exacting maximum impact of 1 (Figure 8). Two pathways (Aminoacy-tRNA biosynthesis, and porphyrin and chlorophyll metabolism) were also observed to be suppressed during tuber endodormancy dormancy stages. Generally, the seven significantly induced pathways can be grouped into three major functional categories, which include: lipids metabolism, energy and nucleotides metabolism, and amino acids metabolism. Lipids and energy nucleotide metabolisms were majorly involved in yam tuber dormancy-breaking activities and their metabolites were preponderance during tuber dormancy-breaking stages. While amino acids metabolisms dominated the activities during yam tuber endodormancy dormancy stages. Table 2 presents the summary of all induced pathways, their log-fold change and impacts on yam tuber dormancy regulation.

## 4. Discussions

Dormancy in yam tuber is a complex process that is synergistically modulated by molecular processes and environmental signals. Physiologically, during the period of dormancy to sprouting, yam tuber transit into three modes of nutrition; from heterotrophic to autotrophic and back to heterotrophic. Metabolites undoubtedly play essential roles during these processes, since they contribute the raw materials for cellular structure composition, energy supply, biochemical reactions, and signaling which are required for the completion of the physiological process of dormancy induction and breaking [18,28]. Usually, upon dormancy induction some specific metabolites are accumulated and some signals activated to maintain low energy conditions in yam tuber that characterize the dormancy period, whereas during dormancy breaking, varieties of metabolites and metabolisms are activated to provide nutrients and energy required for tuber dormancy breaking [7,45]. However, elucidating the key metabolites and metabolic pathways that regulate yam tuber dormancy, to the best of our knowledge has not been done previously, although the functions of some metabolites in botanical seeds have been established [32,46,47,48]. Our study used untargeted metabolomics profiling to determine some key metabolites and their associated metabolic pathways that regulate yam tuber dormancy.

### 4.1. Efficiency of Metabolomics in Deciphering Molecular Mechanisms in a Biological System

In recent years, metabolomics has been frequently used in the study of seed dormancy [28,46,49]. In *Davidia involucrate*: a dove tree that is common in China, which has very recalcitrant seeds that stay in a dormant state for about four years, metabolome profiling identified 48 differentially accumulated metabolites (DAMs) which were enriched in purine metabolism, pyrimidine metabolism, arginine, and proline metabolism, flavone and flavonol biosynthesis, phenylpropanoid biosynthesis, arginine biosynthesis pathways, as well as key phytohormones, abscisic acid, indole-3 acetic acid and sinapic acid that were implicated in the seed dormancy regulation [32]. In rice, comparative metabolic profiling identified 74 candidate metabolites, among which 29 belong to the ornithine-asparagine-polyamine module and the shikimic acid-tyrosine-tryptamine-phenylalamine-flavonoid module. The result revealed that shikimic acid promoted seed dormancy breaking [18]. In Arabidopsis, metabolomic and transcriptomic profiling identified two distinctive profiles involved in the metabolic regulation of seed dormancy breaking and seedling establishment [35]. The distribution of metabolites in two inbred lines (B73 and Mo17) of maize seeds revealed that the two inbred lines were highly differentiated in their metabolite profiles during dormancy breaking, with regard to amino acids, sugar alcohols, and organic acids accumulation, which determined their dormancy phenotype [50]. In the present study, we performed a comparative metabolomics profiling of tubers of two white yam genotypes at different time points during dormancy to dormancy breaking. We observed that a total of 39 metabolites were differentially accumulated in tubers of the two yam genotypes during the study period. The 39 DAMs were dominated by amino acids and derivatives, alkaloids and alcohols, sugars, nucleotides, amines and polyamines, fatty acids and esters, phenols and benzene derivatives, and organic acids. Therefore, this study demonstrated the efficiency of our metabolomic profiling approach in identifying differentially accumulated metabolites, which revealed the dynamic physiological processes in white yam tubers from dormancy to sprouting.

### 4.2. Differentially Accumulated Metabolites Determined the Phenotypic Variation in Dormancy Duration of the Two White Yam Genotypes

Our data showed genotype-specific variation in differentially accumulated metabolites during tuber dormancy and dormancy break of the two white yam genotypes (*TDr1100873* and *Obiaoturugo*). The tubers of *TDr1100873* had 5 unique DAMs that spread between dormancy and dormancy break periods, and exhibited long dormancy phenotype. The 5 unique DAMs observed in the tubers of *TDr1100873* comprised of terpenoids, amines, phenol, and 2 fatty acids. Among the 5 unique DAMs observed in tubers of *TDr110873*, terpenoid was accumulated at 101-DAPM, amine accumulated at 56-DAPM, while phenol and one of the fatty acids accumulated at 87-DAPM, all these time points, except 101-DAPM constitute the dormant period in *TDr1100873*. *Obiaoturugo* had 7 unique DAMs (nucleotide sugar, L-malic acid, amino alcohol, amine, phytohormone, carboxylic acid, and amino acid and derivatives) that also spread between dormancy and dormancy break periods and exhibited short dormant phenotype. Among the 7 unique DAMs observed in the tubers of *Obiaoturugo*, nucleotide sugar, amino acid, and carboxylic acid were all upregulated at 87-DAPM which is the dormancy breaking point in *Obiaoturugo*. The role of amines in dormancy regulation has been shown to be species-specific. In some plants, they promote dormancy breaking (germination), and in others, they induce dormancy, [46,51,52]. Its accumulation in the tuber of long-dormant phenotype is indicative that it induced or at least functioned to prolong yam tuber dormancy. The study has shown phenol as an antioxidant that participates in the scavenging of reactive oxygen species (ROS) thus catalyzing oxygenation reactions through the formation of metallic complexes, and inhibiting the activities of oxidizing enzymes [53]. Its accumulation at the point when the tubers of short dormant genotype-initiated dormancy breaking this suggests it plays a vital role in the process. Conversely, the presence of unique nucleotide sugar and amino acid (glutamine) that accumulated at 87-DAPM which marked the endodormancy breaking point suggests that these metabolites play some roles in determining early dormancy breaking. It has been demonstrated that nucleotide sugar plays a regulatory role in cell division. A strong correlation was observed between the supply of Glc and the expression of cyclins (*cycD2;1, D3;2, A3;2,* and *B1;2*) [54,55], that the accumulation of nucleotide sugar in the tubers of short dormant phenotype might have triggered the early reactivation of cell division and consequently dormancy break. Similarly, it has been reported that L-Glutamine is a direct precursor of Glutamate, and Glutamate occupies a central position in amino acid metabolism in plants and as well plays key role initiation of dormancy breaking process in seeds [56]. We infer that 2 metabolites (amine and phenol) out of the 5 unique DAMs in *TDr1100873* tubers might be playing some role in maintaining prolonged dormancy duration, whereas, 2 metabolites (nucleotide sugar and amino acid) out of the 7 unique DAMs in *Obiaoturugo* tubers might be playing some roles in early dormancy break. The amine and phenol might have induced and maintained dormancy through their antioxidative ability. Since dormancy breaking process generally generates ROS, and ROS has been reported to promote dormancy breaking by oxidation of NADP and release of energy (ATP) required for dormancy breaking process [57] It can be hypothesized that any process that suppresses any process that generates energy in the tuber during dormancy will enhance dormancy maintenance. Conversely, nucleotide sugar and amino acid might promote dormancy break by enhancing energy availability and increasing the nitrogen/carbon ratio which are the primary requirement for dormancy break. Our results agreed with the finding of [58] who reported that composition and changes in concentration of metabolites in the seeds of two wild barley. Our study buttresses the fact that variation in the duration of tuber dormancy could be driven by the absence of nucleotide sugar and nucleic amino acid (L-Glutamine) in the tubers of long dormancy duration genotype (*TDr1100873*).

### 4.3. Metabolic Regulation of Yam Tuber Dormancy

Using three typical methods; ANOVA, PLS-DA, and AHC, for comparative metabolome analysis at different time points during tuber dormancy of the two white yam genotypes, we identified 39 candidates metabolites associated with tuber dormancy regulation, distributed across 14 well-known dormancy regulating chemical functional groups such as: amines and biogenic polyamines, amino acids and derivatives, carboxylic acids, sugars, Alkaloids, alcohols and polyols, flavonoids, phenols, and benzene derivatives, nucleotides, fatty acids and esters, lipids, coumarins and terpenoids, and phytohormones [7,26,59,60]. We did not identify any of the two established dormancy-regulating phytohormones; Abscisic acid (ABA) and Gibberellic acids (GA) directly, although, their precursors were observed in some of the metabolic pathways enriched. Our data showed the dynamic changes in the concentrations of the identified DAMs across the studied dormancy stages of the tubers of the two white yam genotypes in response to different physiological processes that characterized each of the studied dormancy progression stages. Metabolites set enrichment analysis revealed that these DAMs were involved in 12 significantly induced metabolisms during the white yam tuber dormancy-to-dormancy break period. These observations can be further grouped into amines and amino acids metabolisms, secondary metabolites metabolisms, energy metabolisms, and nucleotides metabolisms.

### 4.4. Amines and Amino Acids Metabolisms

Generally, most the amines and amino acids accumulated in the tubers of both yam genotypes during dormant stages, except for L-Glutamine that accumulated during the dormancy breaking stage in the tubers of the short dormant genotype. The preponderance of amines and amino acids during the yam tuber dormancy stage is an indication that the cells of yam tubers during dormancy were in the post-replication phase, implying that replication which is the hallmark of the cell cycle and consequently growth process was halted during this period. It has been reported that active dormancy induction is characterized by the arrest of the cell cycle process at the G1 phase [54]. Studies have also revealed that amino acids have prominent functions in plant growth and development, besides their role as a protein building block, they as well play pivotal roles in the biosynthesis of secondary metabolites, and during signaling processes in plant stress response [61,62]. These functions of amino acids, amines, and polyamines have been shown to be specie-specific, in some plant they promote dormancy breaking (germination), and in others, they induce dormancy [52,63]. The molecular mechanism of exerting any of the dormancy regulatory action (either induction or inhibition) differs too. In dormancy induction, it seems that amino acids and amines utilize the same molecular module that they use when they act as osmo-protectant during plant abiotic and biotic stress response and biosynthesis of secondary metabolites. It has been demonstrated that amino acids have antioxidant properties and act as reactive oxygen species (ROS) scavenger during plant response to stressful conditions [64]. ROS also serve as signaling molecules during plant growth and in the development and responses to biotic and abiotic stresses [65]. ROS had been shown to directly act on plant cell-wall polysaccharides by breaking their glycosidic bonds, which cause the cell wall to relax, thereby promoting cell division and elongation, which marks the breaking of endodormancy, [66]. During seed germination of dicotyledonous plants, such as cress and lettuce, ROS accumulation in their micropylar endosperm and radicle, led to endosperm weakening and radicle elongation, thereby promoting seed germination [67]. Certain cell-wall hydrolases and non-enzymatic substances, such as ROS and expansion, are needed for these processes [68]. Li et al. [57] have shown that ROS accumulated in the radicle and coleorhiza of rice seeds during germination and that ROS production and the number of germinated seeds decreased in the presence of ROS inhibitors, i.e., diphenyleneiodonium chloride and guazatine. We therefore infer that the ROS inhibition action of amino acids in yam tuber led to dormancy induction and maintenance, and hence they accumulated during tuber dormancy in the two-genotype studied. Previous studies have also made similar observations, in soybean (*Glaycin max*), high seed-specific expression of feedback-insensitive genes encoding dihydrodipicolinate synthase (DHPS) and aspartate kinase (AK) resulted in wrinkled seeds and low germination rates [69], while grain yield reduction was observed in high lysine crops [70]. Studies suggest that lysine also affect starch synthesis and endosperm development in maize (*Zea mays* Linn) and rice (*Orayze sative*) [70,71]. It mobilizes sucrose and other energy sources in a cell thereby leading to a state of low-energy condition that characterizes dormancy period. Also, seed mobilization of minor proteins such as seed maturation and dormancy-associated proteins that are degraded during imbibition to provide primary amino acids during germination have been reported [72,73]. In red rice, pyruvate in arginine-proline metabolism was induced in dormant seeds, and implicated in seed dormancy induction and maintenance [74]. In the Dove tree (*D. involucrata Baill*), the seeds that do not germinate after stratification treatment had the strongest pyruvate– arginine– proline metabolism induction, suggesting that this metabolic pathway might be a key contributor to dormancy induction [32]. In strawberries, polyamine in ABA-dominated, IAA, and ethylene-participating crosstalk inhibited seed ripening and germination [75]. In the present study, our pathway topology analysis also revealed that arginine-proline was significantly induced during yam tuber dormancy (Table 2). Our pathway topology analysis also revealed that lysine metabolism leads to the biosynthesis of secondary metabolites such as; tropane, piperidine, and pyridine alkaloids, which are known for dormancy induction and maintenance in the seed. 

Polyamine (putrescine) has also been implicated in other plant physiological processes such as senescence induction, and stress tolerance [76,77]. It has been reported that senescence induction in yam crops is the physiological marker of dormancy onset [7], hence, it is reasonable to hypothesize that putrescine can induce tuber dormancy through the same molecular module that it uses to induce senescence. In the current study, putrescine accumulated in the tubers of both yam genotypes during the dormancy stage, and was suppressed as dormancy progresses towards breaking (germination). Phenylalanine is another important amine that accumulated during dormancy in the present study. Although, the aromatic amine acts as a nitrogen source participates in the activation of cellular processes that induce germination, but its metabolic pathway is also a hub for the biosynthesis of diverse secondary metabolites, most of which counter its own growth-inducing action [78,79]. The study has also demonstrated that upregulation of alanine aminotransferase (*AlaAT*) isoenzymes encoded by long and short dormancy alleles differ in barley seed embryos led to the exhibition of different dormant phenotypes [80] The reduced dormancy allele *Qsd1* mutant enhanced early and uniform germination, whereas long dormancy allele *qsd1* mutant exhibited prolonged dormancy. In the current study, pathway topology analysis revealed that phenylalanine accumulated during tuber dormancy stages, suggesting that the metabolite plays role in tuber dormancy induction. Our pathway topology analysis also revealed that phenylalanine was significantly induced during yam tuber dormancy, and it exerted a significant impact on yam tuber dormancy regulation. 

Our result contrasted several studies that have demonstrated that the majority of amino acids and amines promote dormancy break (germination), and inhibit dormancy induction and maintenance in botanical seeds [52,81,82]. Ornithine and arginine are both nitrogenous amino acids that can form polyamines through decarboxylation [18]. In canola (*Brassica napus*), it was demonstrated that tyramine differentially accumulated in osmoprimed seeds under saline stress, and led to early seed germination [83]. Llebres, et al. [84] identified the genes involved in arginine metabolism in *Pinus pinaster* Ait and found that arginine plays important role in seed germination in *Pinus pinaster*. In cotton, nitrogen flow occurs in the seed during seed formation and germination through asparagine —> arginine —> storage protein —> arginine —> asparagine sequence [85]. The shikimic acid–tyrosine–tryptamine–phenylalanine–flavonoid module is closely related to the pathway of phenylalanine, tyrosine, and tryptophan biosynthesis along with biosynthesis of phenylpropanoids. This module is also well known for its involvement in plant defense systems, especially shikimic acid (shikimate). Some of the members of this pathway have been reported to be involved in seed germination. In legume seeds, tryptamine was found to be the main biogenic amine detected, and its concentration considerably increased during the germination process [86], suggesting that it might be playing role in dormancy breaking. These aforementioned findings contradict the present result, because tyramine accumulated during the yam tuber deep dormancy stage in both genotypes, suggesting that it might be playing role in yam tuber dormancy induction. We also identified one amino acid (sarcosine) that accumulated during yam tuber dormancy, but has not been reported to have any role in plant growth regulation.

Our result demonstrated that amines and amino acids’ role in dormancy induction and maintenance in yam tuber is due to their antioxidant property, and the action of secondary metabolites synthesized in their biosynthetic pathways (Figure 5). The contrast between our result and some of the previous reports of amines and amino acids actions in botanical seeds dormancy regulation could possibly be explained by two reasons namely; (1) The presence of seed coat and requirement of imbibition for dormancy break in botanical seeds, and non-requirement of imbibition for yam tuber dormancy break. In addition to developmental pathways in zygotic tissues, the seed coat plays a critical role in the control of dormancy in botanical seeds of many species. In some seeds, dormancy is known as coat imposed, and dormancy can be removed or reduced by simple removal of or damage to the seed coat. In various species the seed coat has been shown to restrict germination by regulating permeability to either water [87,88], oxygen [89], or germination inhibitors that leach from the seed coat into the embryo [90]. It has been reported that aromatic amines, and amino acids reduced the thickness and increased seed tegument porosity which helped in water imbibition and boost seed germination rate [91]. whereas in the yam tuber, the reverse is the case, as we earlier reported that yam tuber required moisture reduction to a critical threshold of 40% before the dormancy-breaking process could be initiated [44]. The study has also demonstrated that phenolics, aromatic amines, and amino acids in exerting their antioxidant property inhibit enzymes that generate ROS in a cell system [57], in doing this, they might also inhibit enzymes that catalyze the removal of cell cycle arrest at G1 phase, thereby preventing the resumption of mitotic cell division which mark endodormancy break. (2) The botanical seed has a well-organized embryonic system in which dormancy regulation is processed, whereas, yam tuber has a continuous layer of meristematic cells beneath the dried tuber skin in which tuber dormancy regulation is processed, and the basis of yam tuber dormancy on this structure has been explained by Nwogha, et al. [7]. These postulations however need to be further validated through more 

### 4.5. Energy Metabolism

It is believed that yam tuber dormancy induction is an adaptive response to low energy conditions due to the stoppage of the supply of sugar from photosynthesis as a result of vine senescence at tuber physiological maturity [44]. Since photosynthetic sugar supply and mineral uptake systems are not active during yam tuber dormancy, the energy needs for low and high physiological activities during tuber dormancy are supplied from reserved energy-source metabolites in the tuber [45]. In the present study, several energy source metabolites such as; (sucrose, galactose, carboxylic acids, malic acid, fatty acids, esters, and lipids) accumulated at different stages of tuber dormancy. The role of sugar signaling in dormancy regulation has been reported [92]. Based on changes in carbohydrate metabolism, it was suggested that sugar signaling is involved in regulating bud dormancy in non-woody perennial specie of leafy spurge [93]. In woody perennials, bud dormancy is associated with gene expression patterns in the typical manner of carbon starvation, while bud burst in spring depends on sucrose availability, and carbohydrate import from a distant wood parenchyma [94,95]. Expression of an Arabidopsis sucrose-phosphate synthase gene resulted in earlier bud dormancy break in spring transgenic and hybrid poplar (*Populus alba X Populus grandidentata*) [96]. Wengler [92], reported that carbohydrate mobilization for bud dormancy break may depend on interactions between the clock and sugar signaling. While in rice seeds, the study demonstrated that the amount of total storable sugar does not affect the seed dormancy duration, but a mixture of sucrose and raffinose, and thus hypothesized that the ratio of oligosaccharides to sucrose may be the determinant of seed dormancy duration [49,97]. In our study, sucrose accumulated in the tubers of the short dormant genotype (*Obiaoturugo*) at 56-DAPM, whereas, it accumulated in the tubers of the long dormancy duration genotype (*TDr1100873*) at 115-DAP. The 56-DAPM represents 28 days before the appearance of the shoot bud in the short dormant genotype (*Obiaoturugo*), whereas, 115-DAPM was the point of appearance of the shoot bud in the long-dormant genotype (*TDr1100873*). This is an indication that the stored starch might have been mobilized into sucrose at the onset of the dormancy breaking process to provide the required energy to reactivate cellular activities towards tuber dormancy break. our result is in agreement with the findings of [94,95] and the report of Wengler; [93]. Also, we earlier reported that biochemical profiling of non-structural sugar from yam tuber dormancy to dormancy break revealed that the quantity of nonreducing sugar which includes sucrose increased up to 9 and 10 folds in the tubers of *TDr110073* and *Obiaoturugo* respectively and that this increase started from 56-DAPM in tubers of *Obiaoturugo* and 101-DAPM in tubers of *TDr1100873* [44]. This indicates that sucrose plays a role in the early cellular activities culminating in dormancy break in the tubers of both genotypes.

The tricarboxylic acid (TCA) cycle intermediates, L-malic acid was observed in the tubers of *Obiaoturugo* (the short dormancy duration genotype), but not in the tubers of *TDr110873*, and surprisingly, it accumulated at 42-DAPM only. As dormancy progressed towards breaking L-malic acid was downregulated to an almost non-existence level. However, TCA cycles precursors and intermediates such as, carboxylic acids, esters, lipids and fatty acids accumulated from 28 days before the appearance of shoot bud to dormancy break stage in the tubers of both genotypes. This indicates that the TCA cycle plays an essential role in energy metabolism during yam tuber dormancy breaking. Our pathway topology analysis also revealed that the TCA cycle pathway was induced during the yam tuber dormancy progression stage (Table 2). The TCA cycle is one of the central energy metabolic pathways in a biological system, as well as being the biosynthetic pathway of many metabolites [98]. The plant TCA cycle has a set of eight enzymes primarily linking to the oxidation of pyruvate and malate (generated in the cytosol) to CO_2_ with the generation of nicotinamide adenine dinucleotide reduced form (NADH) for oxidation by mitochondrial respiratory chain [99], and subsequently releases the energy required to reactivate cell cycle, which had been arrested at G1 phase during dormancy induction. It has been demonstrated that the accumulation of organic acids, particularly TCA cycle intermediates in plants supports many cellular processes, which are species, tissue, and developmental stage specific [100]. In plants, lipids are the major component of cell membranes, and are used as a compact energy source for seed germination [101]. This was further supported by our enrichment and pathway topology analyses, which revealed that linoleic acid (lipid) metabolism was significantly induced with a fold change ratio above 20 (Figure 7), and its pathway exerted a maximum impact of 1 on yam tuber dormancy regulation (Table 2). We therefore, infer that lipids, carboxylic acids, fatty acids, and esters were metabolized through Embden–Meyerhof–Parnas (EMP), glycolysis, pyruvate, and TCA cycle pathways to generate the required energy pool for yam tuber dormancy break. It is noteworthy to mention that linoleic acid metabolism is very essential for white yam tuber dormancy break, and therefore can be a target metabolite in the genetic manipulation of white yam tuber dormancy duration. Previous studies have also demonstrated that pathways in carbohydrate metabolism and ATP production such as; glycolysis, TCA cycle, EMP pathway, and oxidative phosphorylation were accumulated at the end of endodormancy to produce energy for bud sprouting and as well as precursors for amino acids biosynthesis [46,53]. Put together our results demonstrated that sucrose, carboxylic acids, esters, lipids fatty acids with exception of (octadecadienoic acid) provided the energy required for yam tuber dormancy breaking, with lipid metabolism and its pathway exerting the strongest regulatory impact on the yam dormancy. 

### 4.6. Nucleotides Metabolism and Cellular Processes

Four nucleotides comprised of L-Glutamine (nucleic amino acid), Adenine, nucleotide-sugar, and sugar acid were identified as part of DAMs in our study. Two of these nucleotides (L-Glutamine and nucleotide sugar) were found only in the short dormancy duration genotype (*Obiaoturugo*), whereas, Adenine and sugar acid were found in both genotypes. Adenine accumulated in the tubers of *Obiaoturugo* at 56-DAPM and in the tubers of *TDr1100873* at 101-DAPM. Note; that due to the short dormancy duration of the tubers of *Obiaoturugo,* 56-DAPM in the tubers of *Obiaoturugo* is physiologically equivalent to 101-DAPM in the tubers of *TDr1100873*. Hence, in physiological timing of Adenine accumulation in the tubers of both genotypes is actually the same time, and it marked the point of commencement of molecular activities towards tuber dormancy break in tubers of both genotypes. Hence, the accumulation of adenine in the tubers of both genotypes at their respective stages suggests that adenine plays a key role in the reactivation of cellular processes that led to dormancy break irrespective of genotypic variation. Nucleotide sugar and nucleic amino acid (L-Glutamine) all accumulated at 87-DAPM which marked the appearance of shoot bud in the short dormancy duration genotype, suggesting that these two metabolites play a key role in a later cellular processes that hastened dormancy breaking process in the tubers of *Obiaoturugo*. A previous study has demonstrated that L-Glutamine is direct precursor of Glutamate, and Glutamate occupies a central position in amino acid metabolism in plants and as well plays key role initiation of dormancy breaking process in seeds [56]. Glutamate, glutamine, aspartate, and asparagine are the central regulators of carbon/nitrogen metabolism and they interact with multiple metabolic networks to reactivate the cell cycle process [102,103]. Glutamine and glutamate are precursors of nitrogenous compounds in plants, and the production of glutamate from glutamine is a key point in the synthesis of a variety of organic molecules, such as nucleic acids, amino acids, and secondary metabolites [102]. We infer that L-glutamine could be a biomarker for early yam tuber dormancy breaking, and can be a target metabolic candidate in the genetic manipulation of yam tuber for short dormancy duration. 

Two nucleotide sugars (nucleotide sugar and sugar acid) were identified in this study and both of them accumulated at the onset of dormancy breaking stage. Studies have demonstrated the regulatory role of sugar in cell division. A close correlation was observed between the supply of Glc and the expression of cyclins (*cycD2;1, D3;2, A3;2,* and *B1;2*) [54,56]. The D-type cyclins have been reported to act as a sensor of external conditions within the cellular environment, and are associated with cyclin-dependent kinase (CDKA) to regulate cell cycles [104]. It has also been reported that the regulatory effect of Glc on the rate of cell division is primarily due to signaling rather than nutrient availability and energy status, as cell proliferating activity positively correlated with endogenous hexose levels, but not their uptake rate [54,55]. Implying that nucleotide sugar is not functioning as an energy source, but rather as signaling to reactivate and mobilize other molecular machinery towards the dormancy-breaking process. These reports are in agreement with our data which shows that nucleotide sugar accumulated in the tubers of the short dormancy duration genotype (*Obiaoturugo*) at the onset of dormancy-breaking activities. Sugar acid (galactonolactone) is a precursor of a nucleotide amino acid (ascorbate). The plant has evolved an efficient multi-component antioxidant defense system to protect vital molecules from the damage of mitochondrial generated ROS, as a consequence of the activity of respiratory chains [105]. This antioxidant defense system includes both enzymatic and non-enzymatic elements [106]. One of the most important non-enzymatic members of this system is the water-soluble antioxidant, ascorbate [105]. Paradiso, et al. [107] demonstrated that Ascorbate biosynthesis improved wheat seed kernel development and delayed programmed cell death (PCD), thereby delaying senescing by feeding the plant with its immediate precursor L-galactone-γ-lactone (GL). We infer that galactose accumulation at an early stage of cellular activities towards dormancy break in the current study suggests that it plays double roles during the yam tuber dormancy breaking process. (1) it acted as a precursor in the biosynthesis of ascorbate that scavenged ROS generated during the reactivation of the cell cycle. (2) Galactonolactone also promoted growth and protected the immature cells generated during cell division from being oxidized by ROS, which culminated in a dormancy break. Our result showed that adenine accumulated in the tubers of both genotypes at the onset of dormancy break, 42 days, and 45 days before physical germination of tubers of *TD1100873* and *Obiaoturugo* respectively (Figure 5a,b). This indicates that non de novo synthesis of purine from adenine might have taken place, and synthesis of purine positively regulated the release of the G1 phase of the cell cycle that was arrested during yam tuber dormancy. The release of the arrested G1 phase of the cell cycle marks the release of yam tuber endodormancy, and this also coincides with the 40 days plus that has been reported to be the duration between yam tuber endodormancy release and eco-dormancy release [7]. We postulate that adenine is the metabolic marker for yam tuber endo-dormancy break, whereas, L-glutamine and nucleotide sugar are the potential candidates’ metabolites for early tuber dormancy breaking. In yam, these three candidate metabolites hold the promise of metabolic manipulation of yam crops for a desirable tuber dormancy duration. Therefore, we hypothesize that nucleotide sugar (glucose) might have acted as a signal that triggered the accumulation of L-glutamine, which functions together with the accumulated adenine to increase the nitrogen/carbon ratio (a critical factor in energy metabolism and DNA synthesis) in short dormancy duration genotype (*Obiaoturugo*), thereby leading to its tuber earlier dormancy break.

### 4.7. Secondary Metabolites and Phytohormones Metabolism

Secondary metabolites play key roles in plant growth and developmental processes such as; cell division, hormonal regulation, photosynthetic activity, radicals’ homeostasis, modulation of seeds transition to quiescent state at maturity, environmental signals transduction, nutrient mobilization and mineralization, stress tolerance, plant-soil, and environmental interactions [108,109,110]. In the current study, diverse secondary metabolites that accumulated at different yam tuber dormancy progression stages include; flavonoids, coumarins, terpenoids, alkaloids and alcohols, benzoquinone and benzoic derivatives, phenols, and salicylic acid (hormone). In the tubers of long-dormant genotype (*TDr1100873*) the 3 flavonoids that were detected accumulated during tuber dormancy at (42-DAPM, 56-DAPM) and on sprouted tuber (143-DAPM), whereas, in the tubers of short dormant genotype (*Obiaoturugo*), they accumulated from the onset of dormancy break (56-DAPM, 87-DAPM), and on sprouted tuber (101-DAPM). This suggests that in the tubers of the long dormant genotype flavonoids play role in dormancy induction and maintenance, whereas, in the short dormant genotype, they promote dormancy break (germination). This also explained the variation between the two genotypes with respect to tuber dormancy duration. The flavonoids accumulation in the sprouted tubers indicates that, it might be playing a role also in vine elongation. It has been reported that flavonoids play different roles in different plant species, at different plant growth stages, organs, and tissues with regard to growth regulation [111]. Studies have demonstrated that in botanical seeds, flavonoids induce and prolong seed dormancy [28,59,109,112]. It has been shown that flavonoids induce dormancy by inhibiting reactive oxygen species (ROS), through; suppression of singlet oxygen, inhibition of enzymes that generate ROS, such as (cyclooxygenase, lipoxygenase, monooxygenase, and xanthine oxidase), chelation of ions transition which catalyzes ROS production, suppressing the cascade of free-radicals in lipid peroxidation and recycling of other antioxidants such as amino acids and polyamines [113,114,115]. This agreed with our data on the tubers of the long dormant genotype, but varies with data on the tubers of the short dormant genotype. The opposing role of flavonoids in the tubers of the short dormant genotype compared to the long dormant genotype and botanical seeds presents a frontier for exploration of the new role of flavonoids in dormancy regulation. 

Similarly, coumarins also accumulated in the tubers of the long dormant genotype during the dormant stage (87-DAPM), and in the tubers of short dormant during the onset of dormancy breaking stages (56-DAPM and 87-DAPM). This also indicates that coumarins prolonged dormancy in the tubers of the long dormant genotype, but induced dormancy breaking process in the tubers of the short dormant genotype. Studies have demonstrated that coumarins induced both primary and secondary seed dormancy [116,117,118]. It has been reported that coumarins induce dormancy due to their ability to antagonize gibberellins and auxin’s function, while synergizing with abscisic acid (ABA) [117,119,120]. Our result on the tubers of long dormant phenotype with regard to coumarins metabolism is supported by this model. Conversely, it has been reported that plants show different response depending on the species, size, growth stage, and process of cellular respiration and associated-enormous changes in the mitochondria [121]. This may explain the dormancy-breaking effect of coumarin observed on the tubers of the short dormant genotype in the current study. It has been shown that secondary metabolite can also serve as energy source for cellular processes, when degraded [112]. In light of this, we hypothesize that there could be a mechanism in the tubers of the short dormant genotype that inhibits dormancy induction action of secondary metabolites, and subsequently degrade them at the onset of dormancy breaking to provide additional energy source required to fuel the cell cycle reactivation process, thus leading to their accumulation during this period. It also could be a result of differential accumulation quantity, because it has been reported that extremely high quantities of phenolic compounds including coumarins such as; free (1042%), bound (120%), and total phenolic acid content of (741%) in canary grass seed promoted germination [122]. 

Terpenoid was detected only in the tubers of long dormant phenotype, and it accumulated at the onset of dormancy-breaking activities (101-DAPM). The accumulation of terpenes at 101-DAPM is indicative that it might be playing some role in cellular primary metabolism that is involved in cell cycle reactivation. Although, the direct involvement of terpene in dormancy or plant growth regulation is sparsely reported in the literature. However, terpenes play vital roles in many other plants physiological processes, especially in protection against external invaders, and response to environmental signals [110]. In the recent, it has been reported that terpenes also participate in plant primary metabolism [123]. There is also a report that most dormancy-regulating phytohormones belong to one terpenoid group or the other, for instance, gibberellins belong to diterpenoids, brassinolide belongs to triterpenoids, abscisic acid, and strigolactones belong to tetraterpenoids (carotenoids), cytokinin belongs to isoprene and is also known as primary metabolism terpene [110]. In light of this, it is reasonable to postulate that the terpene that accumulated in the tuber of long dormant genotype at the onset of the dormancy break stage in this study might be a precursor of one of these hormones especially the gibberellins and cytokinin which are known for their role in dormancy breaking and promotion of activities towards germination. Benzoquinone and benzene derivatives also accumulated from the onset dormancy-breaking activities to sprouted tubers in both genotypes, an indication that these metabolites are implicated in the yam tuber dormancy-breaking process. Phenol accumulation was recorded only in the tubers of long dormant genotype, at its mid-dormancy stage (87-DAPM). Study has shown that phenol as an antioxidant, also participate in the scavenging of reactive oxygen species (ROS), catalyzing oxygenation reactions through the formation of metallic complexes, and inhibiting the activities of oxidizing enzymes [53], thereby inducing prolonged dormancy. The absence of phenol in the tubers of the short dormant genotype, and its cumulation in the tubers of the long dormant genotype at 87-DAPM, when endodormancy was broken in the short dormant genotype is indicative of the phenol role in yam tuber dormancy elongation. 

In the tubers of the short dormant genotype, the two alkaloids detected in this study accumulated during the dormancy stage at (42-DAPM), whereas, in the tubers of long dormant genotype one (galactosylglycerol) accumulated at the dormant stage (42-DAPM) and in the sprouted tuber. Isopyridoxal accumulated at the dormancy break stage and in the sprouted tuber. This result demonstrates alkaloids play diverse roles in plant growth and developmental process that depend on the species, plant variety, and growth stage. Hence, in the tubers of the short dormant genotype it induced dormancy, and did not play any role in vine development. In the tubers long dormant genotype, galactosylglycerol played role in dormancy induction, and also show to be playing some role in vine development. Five alcohols detected in this study all accumulated during the early dormancy stage in the tubers of both genotypes, although, two of the alcohols accumulated in dormant and sprouted tubers of both genotypes. Our result shows that alcohols, similar to other osmo-protectants, are also involved in yam tuber dormancy induction, whereas the two that accumulated in sprouted yam tubers might have been mobilized to serve as an energy additional source during the early stage of vine elongation. Studies have demonstrated that alcohols improved crops’ tolerance against abiotic stresses, possess antioxidant properties, and promote germination and seedling development of botanical seeds [124,125,126]. The two hormones (salicylic acid and 3-Cresotinic acid) that were identified as DAMs in this study, all accumulated in the tubers of the short dormant genotype during the dormant stage at (42-DAPM), whereas, 3-Cresotinic acid was not found in the tubers of long dormant genotype. Salicylic acid accumulated in its tubers at the dormancy break stage and in the sprouted tuber. Salicylic acid has been classified as one of the phenolics compounds among the secondary metabolites, and has been reported to act in synergy with ABA to induce dormancy, while it functions alone to promote dormancy break [7,122].

### 4.8. Pathways Elucidations

Our pathway topology analysis revealed that linoleic acid has the strongest influence on the yam tuber dormancy regulation process (Table 2). The pathway view shows that it is a linoleic acid biosynthetic process regulated by Cytochrome P450 (*CYP*) genes family. *CYPs* have been implicated in Gibberellic acids (GAs) biosynthesis and response to an environmental signal (light and photoperiod) during dormancy breaking in many botanical seeds [7], and GAs is one of the major plant growth regulatory hormones that promote seed dormancy break, by inhibiting the action abscisic acid (ABA). ABA and GAs are the two major phytohormones that act in an antagonistic manner to each other to induce and break dormancy respectively, in all crops. The ratio of ABA quantity to GAs quantity in seeds or any other plant tissue that has the capability of being dormant has been reported to be the determinant factor of the seed or tissue dormant status [7], and the effect of this factor (ABA/GAs ratio) surpasses the effect of any other factors and molecular processes that regulate dormancy in plant system. In maize, *CYP01A26* was reported to exhibit ent-kaurene oxidase activity which led to increased accumulation of bioactive GAs and consequently result in early seed dormancy break [127,128] Similarly, in rice, 13-hydroxylation pathway that involves the activity of 13-hydroxlases which were coded by *CYP714b1* and *CYP714B2* genes led to early seed germination [129]. We therefore infer that the activities of *CYP1A2*, *CYP2C*, *CYP2J*, *CYP2EI*, and *CYP3A4*, found in linoleic acid metabolic pathway might have led to increasing the biosynthesis of GAs and activities of GAs in the tubers, led to dormancy break. Although, we did not detect GAs in our study, probably because of the difference between metabolomic profiling and hormonal profiling protocols, however, some of its intermediates were detected which indicates its presence. The Cytochrome P450 (*CYP*) genes upregulation of linoleic acid biosynthesis can lead to inhibition of abscisic acids (ABA) dormancy inducing action, and increased the accumulation of gibberellic acids (GAs) which consequently led to yam tuber dormancy break. Hence, we further infer that the linoleic acid pathway is a putative metabolic pathway that can be exploited for genetic manipulation of yam tuber dormancy. Our pathway topology analysis also revealed that the phenylalanine metabolic pathway has a significant impact on the yam tuber dormancy regulation process. The phenylalanine pathway is a phenylalanine, tyrosine, and tryptophan biosynthetic pathway, and was shown to be significantly induced during yam tuber dormancy (Table 2).

The pathway view shows that it is a hub for the biosynthesis of secondary metabolite, amino acid derivatives, and degradation of one ester that accumulated during the yam tuber dormancy breaking stage (Figure 9). The secondary metabolites synthesized in this pathway accumulated during the tuber dormancy stage, thus establishing a crosstalk link between secondary metabolites and amino acids and how their synergetic action leads to induction and prolonging of dormancy in yam tuber. The ester (octanoic acid, 2-phenylethyl ester) that was degraded in this pathway eventually accumulated during the tuber dormancy breaking, implying that in addition to antioxidant property effects of secondary metabolites and amino acids, they might be antagonizing some tuber dormancy break promoting metabolites such as Octanoic acid, 2-phenylethyl ester, whose action promotes dormancy break. This further supports our earlier postulation that dormancy inducing effect of most of the amino acids in yam tuber was partly as a result of secondary metabolites synthesis along their biosynthetic pathways. In plants, β-alanine is important for the synthesis of pantothenate and subsequently coenzyme A, which is an essential coenzyme in lipid and carbohydrate metabolism [130]. Also, glycolysis and TCA cycle have been reported to be linked to alanine aminotransferase during hypoxia induced waterlogging [131], confirming the role alanine metabolic pathway in stress tolerance, which is been linked to dormancy inducing-action. Moreover, in rice, alanine aminotransferase 1 encoded by the *flo12* gene was found to simultaneously regulate carbon and nitrogen metabolism, while the *flo12* mutant exhibited a floury white-core endosperm [132]. A previous study has demonstrated that phenylalanine, tyrosine and tryptophan are not only essential components of protein synthesis, but are also located upstream of a number of growth hormones and secondary metabolites with multiple biological functions and health-promoting properties, such as protection against abiotic and biotic stress [133]. It has been further shown that phenylalanine is required for protein biosynthesis and cell survival during desiccation in botanical seed, and acts as a precursor of a large number of multi-functional secondary metabolites, among which is a principal structural component in the supporting tissues of vascular plants and some algae [134]. Tyrosine has been reported to be the central hub to various specialized metabolic pathways, including vitamin E and plastoquinone which are essential metabolites of plant nutrition and antioxidant synthesis, as well as a precursor of specialized metabolites with diverse physiological functions such as protein, amino acids, attractants, and defense compounds [135]. It has been demonstrated that tryptophan is an essential amino acid in the synthesis of a large number of bioactive molecules, such as auxin, tryptamine derivatives, phytoalexins, indole glucosinolates, and terpenoid indole alkaloids, as well as playing a pivotal role in the regulation of plant growth and development [62,63].

Two energy metabolic pathways (starch and sucrose metabolism and galactose metabolism) were significantly induced during this study (Figure 9b,c). These pathways are biosynthetic pathways of three important metabolites (sucrose, sugar acid, and nucleotide sugar) that were identified as DAMs in this study. Interestingly, these sugars all accumulated at the onset of dormancy breaking stage in the tubers of both genotypes, except for nucleotide sugar which was detected only in the tubers of the short dormant genotype, and has been earlier highlighted as the possible putative candidate metabolite driving the early dormancy break in that genotype. The significant induction and high impact of these pathways on the yam tuber dormancy regulation process support our earlier postulation that accumulation of these sugars at the onset of tuber dormancy breaking is an indication that they serve as an energy source for the reactivation of the cell cycle which marks the breaking of endodormancy in yam tuber. In addition to the biosynthesis of these sugars, these pathways are also involved in the pentose and glucuronate interconversions through UDP-glucose, pentose phosphate pathway, starch mobilization to amylose, galactosamine, D-glucose-6p, treholse-6p, all of which have been reported to play a pivotal role in cell cycle reactivation, including working in synergy with auxin during the dormancy-breaking process, and, as well, serve as signaling molecules [37,107,136]. Alanine, aspartate, and glutamate metabolism and purine metabolism are other two pathways that were induced during our study period in yam tubers. They are mainly involved in the biosynthesis of nucleotides such as adenine, L-glutamine, DNA precursor, DNA, RNA, histidine proteins and pentose phosphate. Adenine and L-glutamine are the two major metabolites in these pathways that were identified as DAMs in our study, and both accumulated at the onset of the dormancy break stage in both genotypes, which indicates that they play a pivotal role in the array of early cellular events towards dormancy break such as; histidine unpacking, DNA synthesis and replication, synthesis of replicated cells components, including house-keeping RNA during cell division, these required central carbon and nitrogen metabolism. L-glutamine was detected only in the tubers of the short dormant genotype, therefore, the induction and impact of its biosynthesis as revealed by pathway topology analysis support our earlier inference that L-glutamine could be a putative metabolic marker determining the early yam tuber dormancy break phenotype. Put together, our results demonstrate that adenine and galactose (sugar acid) might be conserved putative metabolic for yam tuber dormancy break irrespective of dormancy duration phenotype. Meanwhile, L-glutamine and nucleotide sugar are the two putative metabolic markers for yam tuber short dormant phenotype. Hence, alanine-aspartate-glutamine and purine metabolic pathways are some of the pathways that hold promise around the metabolic manipulation of yam tuber dormancy.

## 5. Conclusions 

This result demonstrates that yam tuber dormancy is a variable trait, and the two genotypes used in this study varied in the duration of their tuber dormancy. This variation was driven by the differential accumulation of terpenoid, phenol, amine, and one fatty acid in the tubers of *TDr110873*, and the differential accumulation of nucleotide sugar, amino acid (L-glutamine), and one carboxylic acid in the tubers of *Obiaoturugo*. The metabolomics changes from dormancy to sprouting in the tubers of both genotypes were systematically revealed, and a total of 39 differentially accumulated metabolites were identified across the investigated yam tuber dormancy stages of both yam genotypes. Amines and biogenic polyamines, amino acids and derivatives, Alcohols, flavonoids, alkaloids, phenols, esters, coumarins, and phytohormone regulated yam tuber dormancy induction and maintenance, whereas, fatty acids, lipids, nucleotides, carboxylic acids, sugars, terpenoids, benzoquinones, and benzene derivatives regulated dormancy breaking and sprouting process in tubers of both yam genotypes. Twelve metabolisms were significantly enriched during yam tuber dormancy and dormancy break process, and these were involved majorly in six metabolic (linoleic acid, phenylalanine, galactose, starch, and sucrose, purine, and alanine-aspartate-glutamate) pathways that exerted significant impacts on yam tuber dormancy regulation. Among these metabolic pathways, only phenylalanine metabolic regulated tuber dormancy induction, whereas, the rest five (linoleic acid, galactose, starch and sucrose, purine and alanine-aspartate-glutamate) pathways regulated dormancy break process, with linoleic acid metabolic pathway exerting strongest regulatory control. Therefore, the linoleic acid pathway holds a promise and the potential of being a putative metabolic pathway to be targeted in the genetic manipulation of yam tuber dormancy. In addition, metabolites such as; adenine, sucrose, and galactose (sugar acid) exhibited the potential of being conserved putative metabolic markers for yam tuber dormancy break irrespective of genotypic variation with respect to dormancy duration. Our results also demonstrated that nucleotide sugar and glutamine are the two potential metabolic markers for early tuber dormancy break, and therefore determined the dormancy duration phenotypic variation between the tubers of two yam genotypes studied. Comparative to the literature information on botanical seeds’ dormancy metabolic regulation mechanisms, our result demonstrated that yam tubers exhibit deviation, especially with regards to the roles of amino acids, amine and polyamines, and some secondary metabolites in dormancy regulations. Our result provides insight into molecular mechanisms regulating yam tuber dormancy, the basis for further metabolomics investigation of yam tuber dormancy regulation mechanisms and future genetic manipulation of yam tuber dormancy.

## Figures and Tables

**Figure 1 metabolites-13-00610-f001:**
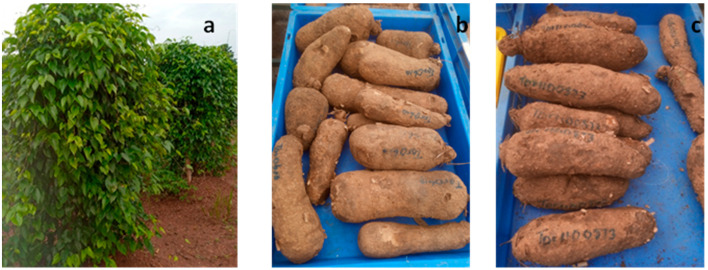
Howing (**a**); established field of the two yam genotypes, (**b**) tubers of *Obiaoturugo* in post-harvest study facility, and (**c**) tubers of *TDr1100873* in post-harvest study facility.

**Figure 2 metabolites-13-00610-f002:**
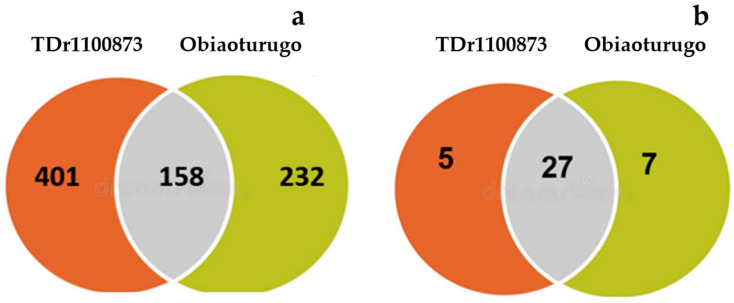
(**a**) Comparison of total annotated metabolites in the tubers of the two yam genotypes. (**b**) Comparison of significantly differentially accumulated metabolites during dormancy in the tubers of the two genotypes.

**Figure 3 metabolites-13-00610-f003:**
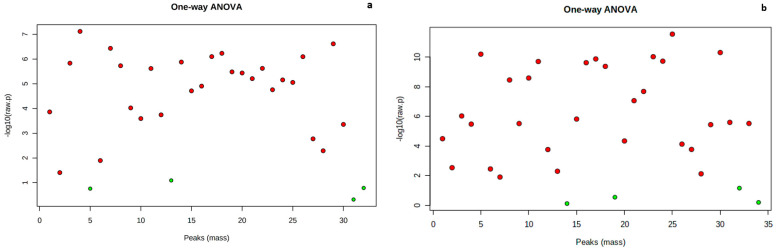
Differential accumulated metabolites identified by ANOVA plot with *p*-value threshold 0.05, FDR ≤ 0.05 and logFC 1.5. Means were compared using Turkey’s Post-hoc test. (**a**) shows DAMs in the tubers of *TDr1100873* across the studied tuber dormancy stages. (**b**) Shows DAMs in the tubers of *Obiaoturugo* across the studied tuber dormancy stages. Red dots are significant differential accumulated metabolites, while the green dots are the metabolites that were differentially accumulated but are not significant. The vertical axis indicates the metabolites log-fold-change, while the horizontal axis shows the mass peak of the metabolites.

**Figure 4 metabolites-13-00610-f004:**
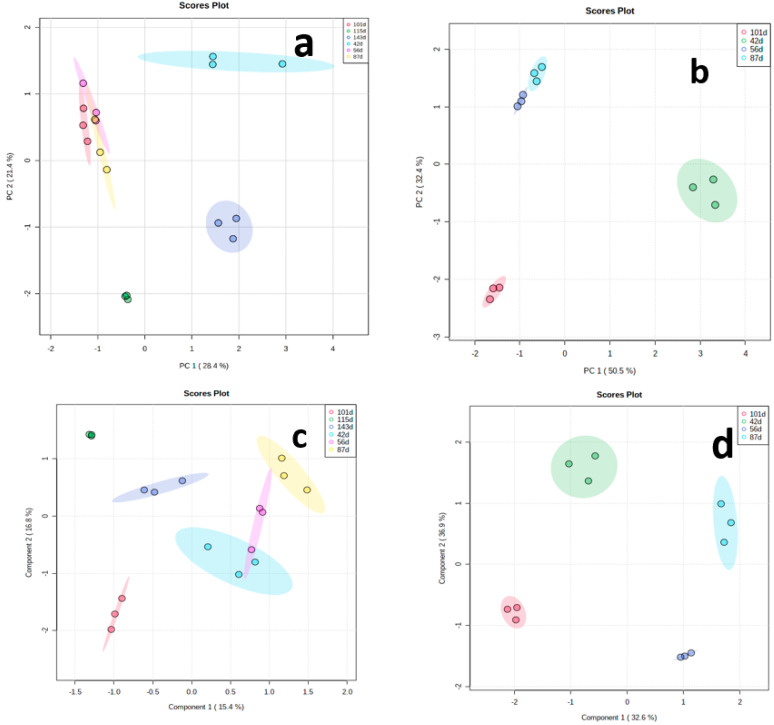
Score plots between the selected Principal components of PCA and components of PLS-DA showing the discrimination of tuber metabolomes of the two genotypes across the dormancy stages studied, and the explained variances are shown in brackets; (**a**) shows the PCA of *TDr1100873*, (**b**) shows the PCA of *Obiaoturugo*, (**c**) shows the PLS-DA of TDr110873, while (**d**) shows the PLS-DA of *Obiaoturugo*.

**Figure 5 metabolites-13-00610-f005:**
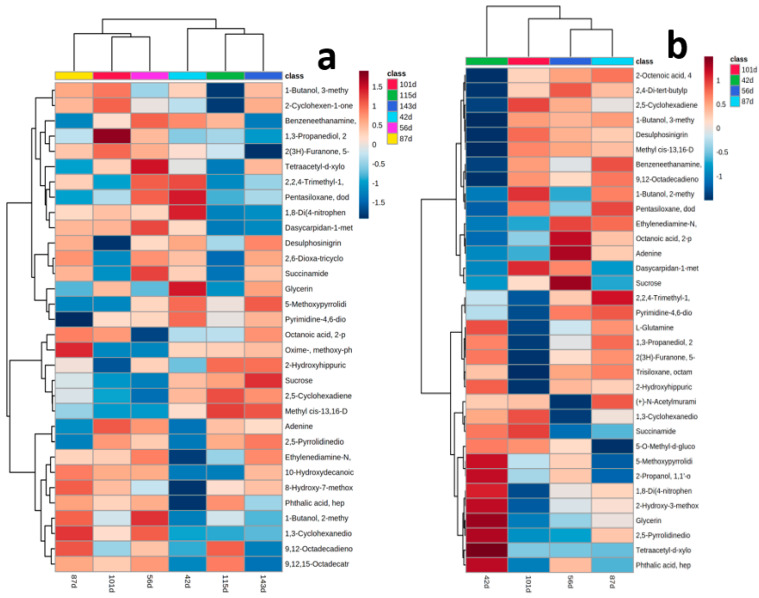
Agglomerative Hierarchical Cluster heatmap of DAMs across the studied dormancy stages in each of the genotypes. Metabolites and dormancy stages were separated into clusters using distance measure Euclidean distance measure, and clustering algorithms ward. linkage. The relative concentration of each metabolite at different studied dormancy stages was standardized, ranked using Spearman’s rank correlation, and shown in the box according to the color criterion indicating their log fold change in concentration at each dormancy stage. The log fold change values according to the color criterion and dormancy stages are shown on the legends at the right of the figures. (**a**) shows the hierarchical cluster heatmap of DAMs in the tubers of *TDr1100873* across the studied dormancy stages, while (**b**) shows the hierarchical cluster heatmap of DAMs in the tubers of *Obiaoturugo* across the studied dormancy stages.

**Figure 6 metabolites-13-00610-f006:**
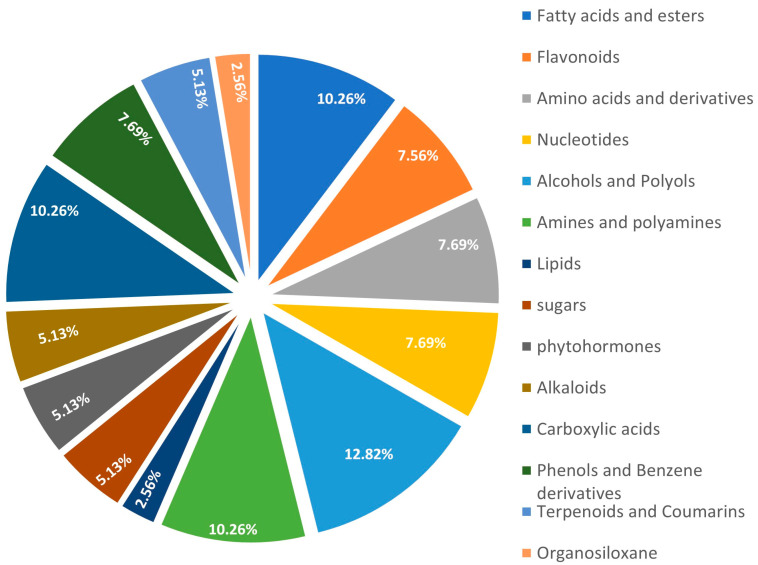
Proportional composition of chemical functional groups of the DAMs. The color and area of each component of the pie chart show the percentage composition of a particular chemical functional group.

**Figure 7 metabolites-13-00610-f007:**
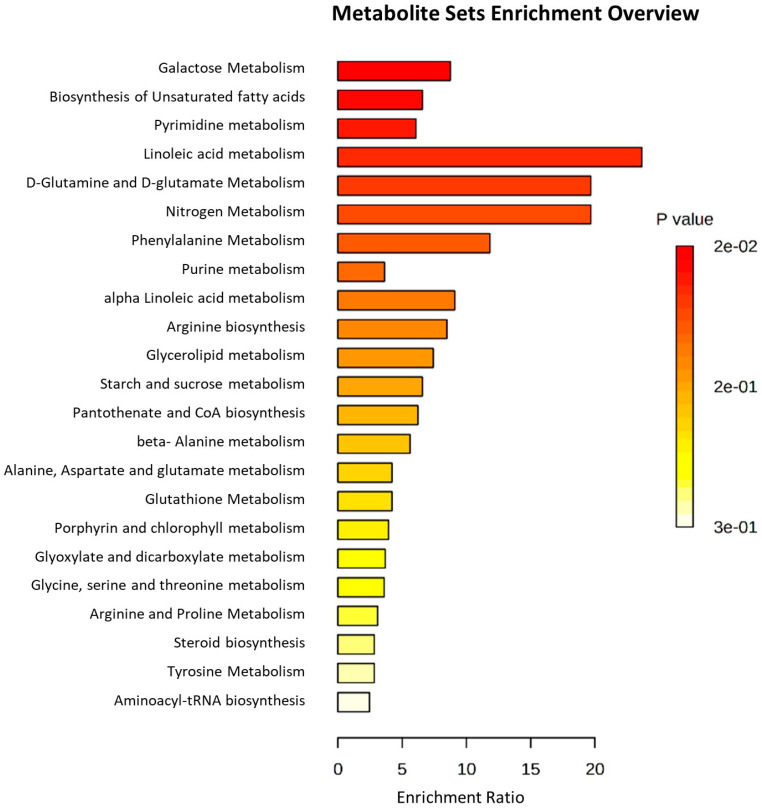
Bar plots for metabolites set enrichment analysis showing the top 23 metabolisms of enriched metabolites. The enrichment ratio values are the ratio between the fold change in the metabolisms across the studied dormancy stages and their expected values under normal conditions. The color criterion indicates the *p*-value at which each metabolism was induced to its enrichment ratio, and it determines whether enrichment is significant or not.

**Figure 8 metabolites-13-00610-f008:**
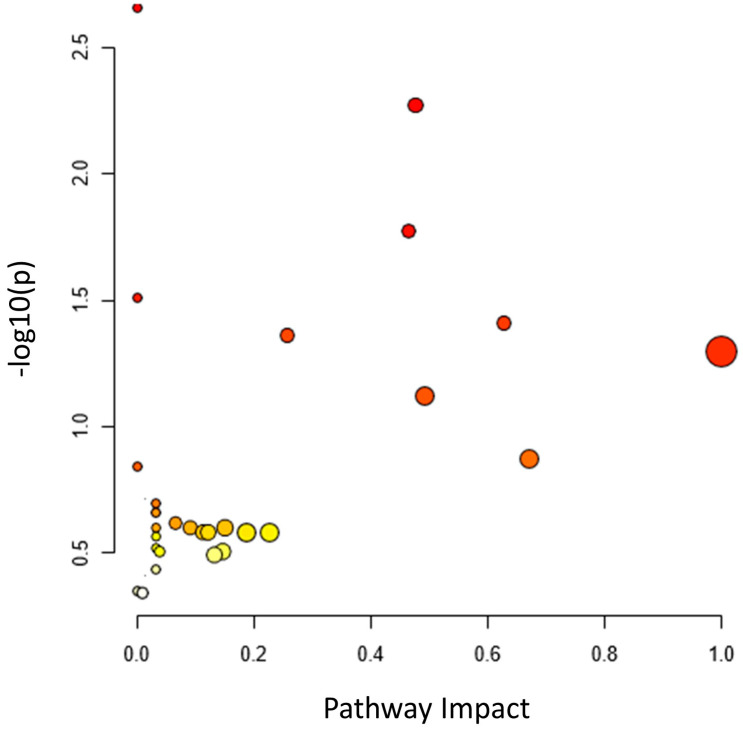
Metabolome view of pathway topology showing the pathways that significantly impacted yam tuber dormancy regulation. The dots denote metabolic pathways, size of the dots indicates the value of the impact of the pathway on yam tuber dormancy regulation on a scale of 0 to 1, whereas, the color intensity of the dots indicates the fold change of the pathway.

**Figure 9 metabolites-13-00610-f009:**
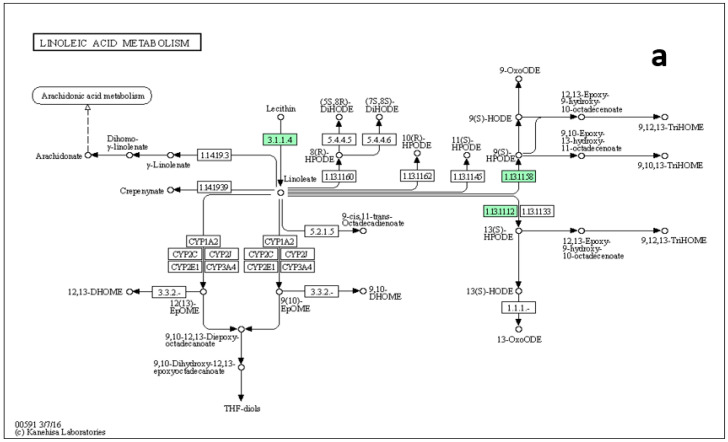
Six candidate pathways identified by pathway topology analysis that have a significant impact on yam tuber dormancy regulation. (**a**) is linoleic acid metabolic pathway, it exerted a maximum impact of 1, on a scale of 0 to 1. (**b**) is phenylalanine metabolic pathway, it exerted an impact of 0.79. (**c**) is galactose metabolic pathway with an impact value of 0.76. (**d**) is starch and sucrose metabolic pathway with an impact value of 0.59. (**e**) is Alanine–aspartate–glutamine metabolic pathway with an impact value of 0.57. (**f**) is the purine metabolic pathway with an impact value of 0.54.

**Table 1 metabolites-13-00610-t001:** Result of mapping of DAMs in HMDB, PubChem, and KEGG for standard name, chemical functional group and IDs determination.

Query	Match	Chem-Fun Group	HMDB	Pub Chem	KEGG
9,12,15-Octadecatrienoic acid, (Z,Z,Z)-	Alpha-Linolenic acid	Fatty acids	HMDB0001388	5280934	C06427
Ethylenediamine-N,N′-dipropionic acid	Edetic Acid	Flavonoids	HMDB0015109	6049	C00284
1-Butanol, 2-methyl-, acetate	Methyl methacrylate	Esters	HMDB0032385	6658	C19504
Tetraacetyl-d-xylonic nitrile	Sarcosine	Amino Acids and Derivatives	HMDB0000271	1088	C00213
(+)-N-Acetylmuramic acid	UDP-N-acetylmuraminate	nucleotide-sugar	HMDB0011720	24755495	C01050
1-Butanol, 3-methyl-, formate	3-Methylbutyl formate	Fatty esters	HMDB0034163	8052	C12293
1,3-Propanediol, 2-ethyl-2-(hydroxymethyl)-	1,3-Butanediol	Alcohols and Polyols	HMDB0031320	7896	C20335
1,8-Di(4-nitrophenylmethyl)-3,6-diazahomoadamantan-9-one	Tyramine	Amino Acids and Derivatives	HMDB0000306	5610	C00483
Benzeneethanamine, 2-fluoro-á,3,4-trihydroxy-N-isopropyl-	Phenylethylamine	Amines, Aromatic	HMDB0012275	1001	C05332
Desulphosinigrin	Delphinidin 3-O-sophoroside	Flavonoids		47205615	C16307
Glycerin	Glycerol	Alcohols and Polyols	HMDB0000131	753	C00116
9,12-Octadecadienoic acid (Z,Z)-	Linoleic acid	Lipids	HMDB0000673	5280450	C01595
Dasycarpidan-1-methanol, acetate (ester)	Acetaminophen	Flavonoids	HMDB0001859	1983	C06804
2,2,4-Trimethyl-1,3-pentanediol diisobutyrate	3-Phenylpropyl 2-methylpropanoate	Alcohols	HMDB0034472	7662	C02008
Sucrose	Sucrose	Disaccharides	HMDB0000258	5988	C00089
5-O-Methyl-d-gluconic acid dimethylamide	3-Cresotinic acid	Phytohormone	HMDB0002390	6738	C14088
2,6-Dioxa-tricyclo [3.3.2.0(3,7)] decan-9-one	Ibuprofen	Amines	HMDB0001925	3672	C01588
1,3-Cyclohexanedione, 2,5,5-trimethyl-	1,4-Cyclohexanedione	Alcohols		10263	C08063
5-Methoxypyrrolidin-2-one	Galactosylglycerol	Alkaloids	HMDB0006790	656504	C05401
2(3H)-Furanone, 5-heptyldihydro	Galactonolactone	Sugar Acids	HMDB0002541	5640	C03383
2-Propanol, 1,1′-oxybis-	Propranolol	Amino Alcohols	HMDB0001849	4946	C07407
10-Hydroxydecanoic acid	12-Hydroxydodecanoic acid	Fatty Acids	HMDB0002059	79034	C08317
Succinamide	Putrescine	Biogenic Polyamines	HMDB0001414	1045	C00134
2-Octenoic acid, 4,5,7-trhydroxy	Methyl acrylate	Carboxylic Acids	HMDB0033977	7294	C19443
2-Hydroxy-3-methoxy-succinic acid, dimethyl ester	L-Malic acid	carboxylic acid	HMDB0000156	222656	C00149
Octanoic acid, 2-phenylethyl ester	Styrene	Benzene Derivatives	HMDB0034240	7501	C19506
2-Cyclohexen-1-one, 4-(3-hydroxy-1-butenyl)-3,5,5-trimethyl-	Amoxycillin	Terpenoids	HMDB0030500	2171	C06827
8-Hydroxy-7-methoxycoumarin	Oxybenzone	Coumarins	HMDB0015497	4632	C14285
Pyrimidine-4,6-diol, 5-methyl-	Uracil	Nucleic Acids and Derivatives	HMDB0000300	1174	C00106
Oxime-, methoxy-phenyl-_	3,4-Dihydroxyphenylacetaldehyde	phenols	HMDB0003791	119219	C04043
Adenine	Adenine	Nucleotides	HMDB0000034	190	C00147
2,5-Cyclohexadiene-1,4-dione, dioxime	Quinone	Benzoquinones	HMDB0003364	4650	C00472
2,5-Pyrrolidinedione, 3-(1-aminoethylidene)-4-methyl-	Isopyridoxal	Alkaloids	HMDB0004290	440899	C06051
Phthalic acid, hept-4-yl isobutyl ester	Thalidomide	Carboxylic Acids	HMDB0015175	92142	C07910
L-Glutamine	L-Glutamine	Amino Acids and Derivatives	HMDB0000641	5961	C00064
Methyl cis-13,16-Docosadienate	Methylitaconate	carboxylic acid	METPA0268		C02295
2-Hydroxyhippuric acid-3TMS	Salicyluric acid	Phytohormone	HMDB0000840	10253	C07588
2,4-Di-tert-butylphenoxytrimethylsilane	Butenafine	Amines	HMDB0015223	2484	C08067
Pentasiloxane, dodecamethyl-	Dodecamethylpentasiloxane	organosiloxane	HMDB0062731	8853	

**Table 2 metabolites-13-00610-t002:** Result of pathway topology analysis, showing the significantly induced pathways and their impact on yam tuber dormancy regulation.

	Total	Expected	Hits	Raw *p*	−log10 (*p*)	Holm Adjust	FDR	Impact
Isoquinoline alkaloid biosynthesis	6	0.08	2	2.26 × 10^−3^	2.65 × 10	2.15 × 10^−1^	2.03 × 10^−1^	0.00
Galactose metabolism	27	0.35	3	4.28 × 10^−3^	2.37 × 10	4.02 × 10^−1^	2.03 × 10^−1^	0.76
Tyrosine metabolism	18	0.23	2	2.11 × 10^−2^	1.68 × 10	1.00 × 10	6.67 × 10^−1^	0.04
Biosynthesis of unsaturated fatty acids	22	0.28	2	3.09 × 10^−2^	1.51 × 10	1.00 × 10	7.33 × 10^−1^	0.00
Linoleic acid metabolism	4	0.05	1	5.04 × 10^−2^	1.30 × 10	1.00 × 10	8.15 × 10^−1^	1.00
Glyoxylate and dicarboxylate	29	0.37	2	5.14 × 10^−2^	1.29 × 10	1.00 × 10	8.15 × 10^−1^	0.06
metabolism								
Pyrimidine metabolism	38	0.49	2	8.32 × 10^−2^	1.08 × 10	1.00 × 10	1.00 × 10	0.06
Lysine biosynthesis	9	0.12	1	1.10 × 10^−1^	9.59 × 10^−1^	1.00 × 10	1.00 × 10	0.47
Nitrogen metabolism	12	0.15	1	1.44 × 10^−1^	8.41 × 10^−1^	1.00 × 10	1.00 × 10	0.54
Phenylalanine metabolism	12	0.15	1	1.44 × 10^−1^	8.41 × 10^−1^	1.00 × 10	1.00 × 10	0.79
Purine metabolism	63	0.81	2	1.92 × 10^−1^	7.17 × 10^−1^	1.00 × 10	1.00 × 10	0.26
Arginine biosynthesis	18	0.23	1	2.09 × 10^−1^	6.81 × 10^−1^	1.00 × 10	1.00 × 10	0.00
beta-Alanine metabolism	18	0.23	1	2.09 × 10^−1^	6.81 × 10^−1^	1.00 × 10	1.00 × 10	0.00
Citrate cycle (TCA cycle)	20	0.26	1	2.29 × 10^−1^	6.40 × 10^−1^	1.00 × 10	1.00 × 10	0.33
Zeatin biosynthesis	21	0.27	1	2.39 × 10^−1^	6.22 × 10^−1^	1.00 × 10	1.00 × 10	0.00
Carbon fixation in photosynthetic organ-	21	0.27	1	2.39 × 10^−1^	6.22 × 10^−1^	1.00 × 10	1.00 × 10	0.06
isms								
Glycerolipid metabolism	21	0.27	1	2.39 × 10^−1^	6.22 × 10^−1^	1.00 × 10	1.00 × 10	0.26
Phenylalanine, tyrosine, and tryptophan	22	0.28	1	2.49 × 10^−1^	6.04 × 10^−1^	1.00 × 10	1.00 × 10	0.08
biosynthesis								
Starch and sucrose metabolism	22	0.28	1	2.49 × 10^−1^	6.04 × 10^−1^	1.00 × 10	1.00 × 10	0.59
Pyruvate metabolism	22	0.28	1	2.49 × 10^−1^	6.04 × 10^−1^	1.00 × 10	1.00 × 10	0.24
Alanine, aspartate and glutamate	22	0.28	1	2.49 × 10^−1^	6.04 × 10^−1^	1.00 × 10	1.00 × 10	0.57
metabolism								
Pantothenate and CoA biosynthesis	23	0.30	1	2.59 × 10^−1^	5.87 × 10^−1^	1.00 × 10	1.00 × 10	0.00
Cyanoamino acid metabolism	26	0.33	1	2.87 × 10^−1^	5.42 × 10^−1^	1.00 × 10	1.00 × 10	0.00
Glutathione metabolism	27	0.35	1	2.97 × 10^−1^	5.28 × 10^−1^	1.00 × 10	1.00 × 10	0.01
alpha-Linolenic acid metabolism	27	0.35	1	2.97 × 10^−1^	5.28 × 10^−1^	1.00 × 10	1.00 × 10	0.11
Arginine and proline metabolism	28	0.36	1	3.06 × 10^−1^	5.14 × 10^−1^	1.00 × 10	1.00 × 10	0.16
Glycine, serine, and threonine metabolism	33	0.42	1	3.50 × 10^−1^	4.56 × 10^−1^	1.00 × 10	1.00 × 10	0.00
Aminoacyl-tRNA biosynthesis	46	0.59	1	4.53 × 10^−1^	3.44 × 10^−1^	1.00 × 10	1.00 × 10	0.00
Porphyrin and chlorophyll metabolism	47	0.60	1	4.61 × 10^−1^	3.37 × 10^−1^	1.00 × 10	1.00 × 10	0.01

**Total** is the number of compounds in the pathway, **expected** is the normal ratio value, **hits** are the matched number from the uploaded data, **Raw *p*** is the *p*-value before adjustment, **Holm *p*** is the *p*-value adjusted using the Holm–Bonferroni method. **FDR** is the *p* value adjusted using a false discovering rate. **Impact** is the effect of the pathway on a biological process (yam tuber dormancy regulation).

## Data Availability

The data presented in this study are available in the Appendix A here.

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
