# Peer review of "Comparative Metabolomics Profiling Reveals Key Metabolites and Associated Pathways Regulating Tuber Dormancy in White Yam (Dioscorea rotundata Poir.)"

_metabolites, 2023, doi:10.3390/metabo13050610_

Round 1

Reviewer 1 Report

I have carefully examined the manuscript entitled “Comparative metabolomics Profiling reveals key metabolites and associated pathways regulating tuber dormancy in white yam (Dioscorea rotundata)” by Nwogha et al. In this paper authors describe identification and quantitative analysis, by untargeted GCMS analysis, of the metabolic profiles of two white yam tubers genotypes in different moment of their dormancy cycle. Among the 949 annotated metabolites, 39 differentially accumulated metabolites were connected to the yam dormancy induction and breaking and to the potential metabolic pathways involved in the regulation of yam tuber dormancy. Authors give relevance to their results due to potential application in genetic manipulation for the reduction of yam tuber dormancy, useful to increase annual yam tuber harvesting.

Authors performed a complex and overall correct comparative investigation, collecting a huge amount of data, successively submitted to PCA, PLS and AHC statistical analysis.

I have appreciated authors metabolomic approach to give insights on the dormancy cycle of Dioscorea rotundata tubers, and the following discussion focused to give rational connections between the collected data and the molecular mechanisms regulating tuber dormancy.

Nevertheless, I have some comments and answers to be responded by authors:

a) Paragraph 2.5

This paragraph needs to be revised, for several spelling errors. Is the protocol used for the extraction efficient to cover all the metabolites? Can you prove that the extraction temperature doesn’t interfere with metabolites stability?

b) Paragraph 2.6

Correct some errors in the composition of the GCMS method.

Data processing need to be improved. Author should explain how the data matrix used for PCA and PLS-DA is composed. How many observations and how many variables are included? It is not well clear how the analysis was performed.

In materials and methods authors wrote that samples were collected and analyzed in triplicates but in PCA score scatter plot just 3 sample for each type of clusters are shown.

Moreover, relative figures need to be improved in resolution. Labels are not clear (enhance font size), and figure captions must be improved.

Clarify which criteria have been used for metabolites identification.

Moreover, since 949 metabolites were annotated, a list of them should be reported in SI.

The manuscript is written in a good English language, even if a general revision on punctuation is recommended.

The work could be considered as an interesting contribution to the analysis of plant cycles and their potential modulation.

On this basis, I will consider the above-mentioned manuscript suitable for publication on Metabolites, once the authors will reply to the above comments.

Author Response

On behalf of the co-authors of this manuscript I wish to appreciate the reviewer for his or her critique and inputs which have improved the quality of manuscript.

The following are our responses to the reviewers 1 comments and our responses.

Comment 1: Paragraph 2.5

This paragraph needs to be revised, for several spelling errors. Is the protocol used for the extraction efficient to cover all the metabolites? Can you prove that the extraction temperature doesn’t interfere with metabolites stability?

Response: The revision has been done. On the question about the efficiency of the extraction method, the answer is yes, the method is robust enough and has even be used on metabolies extraction from yam by Price et al 2016 and 2017. 

Comment 2 b) Paragraph 2.6

Correct some errors in the composition of the GCMS method

Response: corrected.

Comment 3; Data processing need to be improved. Author should explain how the data matrix used for PCA and PLS-DA is composed. How many observations and how many variables are included? It is not well clear how the analysis was performed.

Response: Data processing is simply normalization, and we’ve reported the normalization process in the possible most clarity of term and attacked the graphs as supplementary material S1 and 2. Data matrix for PCA and PLS-DA are the mean of the triplicate replicates of each sampling point. The software (metaboanalyst) that was used to perform the analysis has a way generating mean of the replicates of each data point using in-built R script and all you need to do is select the option for mean during the analysis.

Comment 4; In materials and methods authors wrote that samples were collected and analyzed in triplicates but in PCA score scatter plot just 3 sample for each type of clusters are shown.Moreover, relative figures need to be improved in resolution. Labels are not clear (enhance font size), and figure captions must be improved.

Response: The same as above. The PCA and PLS-DA are mean value of each data point. And it describes the metabolic content of tuber at that particular point and how it differs from the content at another sampling point. We have improved the resolution of all the figures, but it was not possible for it to be done while review change track is on as it was posing some challenges.

Comment 5; Clarify which criteria have been used for metabolites identification.

Moreover, since 949 metabolites were annotated, a list of them should be reported in SI

Response: We stated in the relevant section of methodology that deconvolution and peak identification was done using the GC MS machine in-built metabolite libraries which were listed in that same section. The PDF file containing all the metabolites’ peak and annotation was attached as supplementary material during submission.

Once again, I want to sincerely appreciate the efforts of the reviewer towards ensuring that this manuscript comes out in its possible best form.

Yours Sincerely

Jeremiah S. Nwogha

Reviewer 2 Report

Journal Metabolites (ISSN 2218-1989)

Manuscript ID metabolites-2316345

Type Article

Title  Comparative metabolomics Profiling reveals key metabolites and associated pathways regulating tuber dormancy in white yam (Dioscorea rotundata)

Authors Jeremiah S. Nwogha , Abtew G. Wosene , Muthurajan Raveendran , Jude E. Obidiegwu , Happiness O. Oselebe , Rohit Kambale , Cynthia A Chilaka * , Veera Ranjani R

Section Plant Metabolism

Special Issue Plant Metabolic Genetic Engineering

Title

1. Add the abbreviation of the discoverer’s name after the Latin name of the species

Abstract

2. Check that all abbreviations are explained; give the explanation first and then the abbreviation

Keywords

3. Eliminate words that are used in the title

Introduction

4. l. 42 – 58 I suggest adding the percentage content of biologically active chemical compounds (the authors only mention their presence) and indicate their main properties used in many industries based on the latest literature reports closely related to the subject of the study

5. l. 59 – 65 complete the citation

6. l. 47, 60 check the punctuation here and throughout the manuscript

7. l. 66 – 89 the reader expects more information, hence, I suggest adding an explanation of the metabolic processes related to the subject of the study based on the latest literature reports

“Dormancy induction and release is associated with numerous physiological and biochemical processes that are regulated by gene expression, protein synthesis, hormonal signaling and energy metabolism [14].”

8. l. 90 – 105 I suggest adding an explanation of this metabolism, referring to the subject of the study

„This effect has been suspected to involve starch metabolism, phytohormones interactions and antioxidant defenses ...

9. l. 106 – 130 Please reedit the text in order to address to the topic of the study closely.

10. Dioscorea rotundata represents the family Dioscoreaceae; the authors cite various reports, often relating to taxonomically distant plants “..Picea glauca and Julans regia…”, “…longevity in Brassiceae and tomato …”.

11. l. 132 – 146. Please add the rationale behind taking up the topic and a clearly defined aim of the study.

12. Formulate the research theses.

Material and research methods

13. Please check the layout of the manuscript in accordance with the „Experimental Design” requirements for authors.

14. Correct the description of Figure 1 in an adequate way for this type of high-IF journals with international coverage, e.g. eliminate repetitions (similarly throughout the manuscript).

15. l. 148-250. Please add literature references for the methodology in the Material and methods section.

Results

16. l. 252 – 491

Check the citations of all figures and the correctness of their notation, e.g.:

„(Fig. 2a). Differentially accumulated metabolites (DAMs) in tubers of the two genotypes 258 were also compared, and Fig2b”

17. Figures 3a, b; 4a–d; 5a, b, 6, 7 are illegible; please improve the quality and correct the notation: 3a, b instead of 3a -b, similarly 5a-b

Discussions

18. l. 579-597 Please complete the discussion.

19. The discussion is not a place to quote figures and tables from the results, e.g.:

„…pathways (Fig 5)…”

“above 20 (Fig 7), and its pathway exerted maxi-777 mum impact of 1 on yam tuber dormancy regulation (table 2).”

„respectively (Fig 5a-b).”

Conclusions

20. The Conclusions should provide a specific answer to the research theses and the aim of the study and specific conclusions from the research should be formulated.

References

21. The section requires a thorough revision in accordance with the guidelines for authors, e.g. the Latin names of genera and species should be italicized, the notation of the title of the journal should be standardized, etc.; there are so many flaws that it is impossible to list them all.

Author Response

On behalf of the co-authors of this manuscript I wish to appreciate the reviewer for his or her critique and inputs which have improved the quality of manuscript.

The following are our responses to the reviewers 1 comments and our responses.

Title

  1. Add the abbreviation of the discoverer’s name after the Latin name of the species

Response: done

Abstract

  1. Check that all abbreviations are explained; give the explanation first and then the abbreviation

Response: done

Keywords

  1. Eliminate words that are used in the title

Response: Some amendment made

Introduction

  1. l. 42 – 58 I suggest adding the percentage content of biologically active chemical compounds (the authors only mention their presence) and indicate their main properties used in many industries based on the latest literature reports closely related to the subject of the study.

Response: We feel that such details are not necessary since they don’t have direct bearing on our topic.

  1. l. 59 – 65 complete the citation

Response: done

  1. l. 47, 60 check the punctuation here and throughout the manuscript

Response: done

  1. l. 66 – 89 the reader expects more information; hence, I suggest adding an explanation of the metabolic processes related to the subject of the study based on the latest literature reports

Response: To make the introduction as concise as possible, we sieved out the information presented and we felt that such details are not necessary.

“Dormancy induction and release is associated with numerous physiological and biochemical processes that are regulated by gene expression, protein synthesis, hormonal signaling and energy metabolism [14].”

  1. l. 90 – 105 I suggest adding an explanation of this metabolism, referring to the subject of the study

Response: we’ve explained what that are relevant to this topic, besides, this is introduction and not discussion.

  1. l. 106 – 130 Please reedit the text in order to address to the topic of the study closely.

Response; done

  1. Dioscorea rotundatarepresents the family Dioscoreaceae; the authors cite various reports, often relating to taxonomically distant plants “..Picea glauca and Julans regia…”, “…longevity in Brassiceae and tomato …”.

Response: This is pioneer work on yam; therefore, we’ve cited only available reports, besides, dormancy is a very conserved across wide range of species.

  1. l. 132 – 146. Please add the rationale behind taking up the topic and a clearly defined aim of the study.
  2. Formulate the research theses.

Response: 64-74, presented the rationale that informed our decision to take up this research. 135-141, clearly stated the objective of the research, while 142-150 presented the justification.

Material and research methods

  1. Please check the layout of the manuscript in accordance with the „Experimental Design”requirements for authors

Response: done

  1. Correct the description of Figure 1 in an adequate way for this type of high-IF journals with international coverage, e.g. eliminate repetitions (similarly throughout the manuscript).

Response: We feel strongly that there is nothing to correct in figure 1 because it is just a figure showing yam crop during vegetative growth phase, and harvested tuber in a storage facility, it is not result of any sort.

  1. l. 148-250. Please add literature references for the methodology in the Material and methods section

Response: done

Results

  1. l. 252 – 491

Check the citations of all figures and the correctness of their notation, e.g.:

„(Fig. 2a). Differentially accumulated metabolites (DAMs) in tubers of the two genotypes 258 were also compared, and Fig2b”

  1. Figures 3a, b; 4a–d; 5a, b, 6, 7 are illegible; please improve the quality and correct the notation: 3a, b instead of 3a -b, similarly 5a-b

Response: done

Discussions

  1. l. 579-597 Please complete the discussion.

Response; done

  1. The discussion is not a place to quote figures and tables from the results, e.g.:

„…pathways (Fig 5)…”

“above 20 (Fig 7), and its pathway exerted maxi-777 mum impact of 1 on yam tuber dormancy regulation (table 2).”

Response; We feel strongly that it is out of necessity and appropriate, since the pathway elucidation was done only in the discussion.

Conclusions

  1. The Conclusions should provide a specific answer to the research theses and the aim of the study and specific conclusions from the research should be formulated.

Response; We feel strongly that our conclusion satisfied these points.

References

  1. The section requires a thorough revision in accordance with the guidelines for authors, e.g. the Latin names of genera and species should be italicized, the notation of the title of the journal should be standardized, etc.; there are so many flaws that it is impossible to list them all.

Response: We used endnote and MDPI style in generating the references.

Once again, I want to sincerely appreciate the efforts of the reviewer towards ensuring that this manuscript comes out in its possible best form.

Yours Sincerely

Jeremiah S. Nwogha

Reviewer 3 Report

Dear authors, I really appreciate the efforts of all the authors involved. The work done is really of good quality. I have attached my comments (mostly minor) to the attached PDF. Besides them, I have a few major comments:

1. The results and discussions can be written in a more organized way. There are many unnecessary repetitions.

2. Since this work is application-based, I suggest focusing the discussion more on what future applications there may be in this field.

3. Please revise all the figures, currently it appears like a screenshot.

After addressing these comments, I endorse the publication of this manuscript.

Author Response

On behalf of the co-authors of this manuscript I wish to appreciate the reviewer for his or her critique and inputs which have improved the quality of manuscript.

The following are our responses to the reviewers 1 comments and our responses.

Comment 1. The results and discussions can be written in a more organized way. There are many unnecessary repetitions.

Response: The repetitiveness of the results presentation is almost unavoidable since our study is time series in nature and we have looked at the changes in yam tuber metabolomics over time. It is difficult for us not talk about these changes across the data time point that they occurred because we want to avoid repetition.

Comment 2. Since this work is application-based, I suggest focusing the discussion more on what future applications there may be in this field.

Response; Yes, our ultimate aim is to use the apply the knowledge and information generated from this research in genetic or metabolic manipulation of yam tuber dormancy duration. But we have to identify and the describe the molecular mechanism regulating the yam dormancy duration, and that’s what we have done here. The next study will focus on the application of the knowledge generated from here.

Comment 3 Please revise all the figures, currently it appears like a screenshot.

Response: done.

Once again, I want to sincerely appreciate the efforts of the reviewer towards ensuring that this manuscript comes out in its possible best form.

Yours Sincerely

Jeremiah S. Nwogha

Round 2

Reviewer 2 Report

Please do the correction of the references section